# Uncertainty-based Offline Variational Bayesian Reinforcement Learning for Robustness under Diverse Data Corruptions

**Rui Yang**[1,2]**, Jie Wang**[1,2]*****, Guoping Wu**[1]**, Bin Li**[1]
[1]University of Science and Technology of China
[2]MoE Key Laboratory of Brain-inspired Intelligent Perception and Cognition
`{yr0013, guoping}@mail.ustc.edu.cn`
`{jiewangx, binli}@ustc.edu.cn`

## Abstract

Real-world offline datasets are often subject to data corruptions (such as noise or adversarial attacks) due to sensor failures or malicious attacks. Despite advances in robust offline reinforcement learning (RL), existing methods struggle to learn robust agents under high uncertainty caused by the diverse corrupted data (i.e., corrupted states, actions, rewards, and dynamics), leading to performance degradation in clean environments. To tackle this problem, we propose a novel robus**t** va**r**iational **B**a**y**esian inferen**c**e for offlin**e R**L (TRACER). It introduces Bayesian inference for the first time to capture the uncertainty via offline data for robustness against all types of data corruptions. Specifically, TRACER first models all corruptions as the uncertainty in the action-value function. Then, to capture such uncertainty, it uses all offline data as the observations to approximate the posterior distribution of the action-value function under a Bayesian inference framework. An appealing feature of TRACER is that it can distinguish corrupted data from clean data using an entropy-based uncertainty measure, since corrupted data often induces higher uncertainty and entropy. Based on the aforementioned measure, TRACER can regulate the loss associated with corrupted data to reduce its influence, thereby enhancing robustness and performance in clean environments. Experiments demonstrate that TRACER significantly outperforms several state-of-the-art approaches across both individual and simultaneous data corruptions.

## 1 Introduction

Offline reinforcement learning (RL) aims to learn an effective policy from a fixed dataset without direct interaction with the environment [1, 2]. This paradigm has recently attracted much attention in scenarios where real-time data collection is expensive, risky, or impractical, such as in healthcare [3], autonomous driving [4], and industrial automation [5]. Due to the restriction of the dataset, offline RL confronts the challenge of distribution shift between the policy represented in the offline dataset and the policy being learned, which often leads to the overestimation for out-of-distribution (OOD) actions [1, 6, 7]. To address this challenge, one of the promising approaches introduce uncertainty estimation techniques, such as using the ensemble of action-value functions or Bayesian inference to measure the uncertainty of the dynamics model [8–11] or the action-value function [12–15] regarding the rewards and transition dynamics. Therefore, they can constrain the learned policy to remain close to the policy represented in the dataset, guiding the policy to be robust against OOD actions.

---

*Corresponding author. Email: jiewangx@ustc.edu.cn.

38th Conference on Neural Information Processing Systems (NeurIPS 2024).

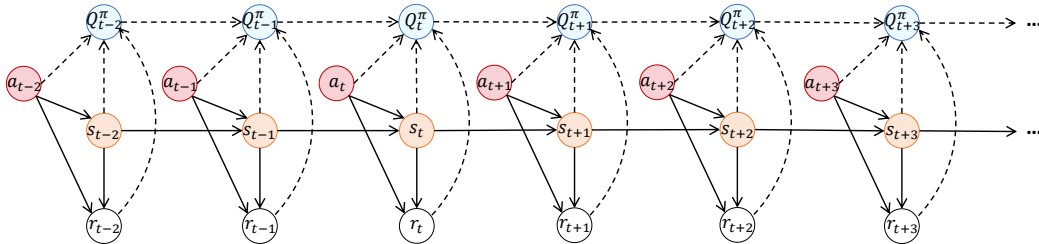

Figure 1: Graphical model of decision-making process. Nodes connected by solid lines denote data points in the offline dataset, while the Q values (i.e., action values) connected by dashed lines are not part of the dataset. These Q values are often objectives that offline algorithms aim to approximate.

Nevertheless, in the real world, the dataset collected by sensors or humans may be subject to extensive and diverse corruptions [16–18], e.g., random noise from sensor failures or adversarial attacks during RLHF data collection. Offline RL methods often assume that the dataset is clean and representative of the environment. Thus, when the data is corrupted, the methods experience performance degradation in the clean environment, as they often constrain policies close to the corrupted data distribution.

Despite advances in robust offline RL [2], these approaches struggle to address the challenges posed by diverse data corruptions [18]. Specifically, many previous methods on robust offline RL aim to enhance the testing-time robustness, learning from clean datasets and defending against attacks during testing [19–21]. However, they cannot exhibit robust performance using offline dataset with perturbations while evaluating the agent in a clean environment. Some related works for data corruptions (also known as *corruption-robust offline RL* methods) introduce statistical robustness and stability certification to improve performance, but they primarily focus on enhancing robustness against adversarial attacks [16, 22, 23]. Other approaches focus on the robustness against both random noise and adversarial attacks, but they often aim to address only corruptions in states, rewards, or transition dynamics [24, 17]. Based on these methods, recent work [18] extends the data corruptions to all four elements in the dataset, including states, actions, rewards, and dynamics. This work demonstrates the superiority of the supervised policy learning scheme [25, 26] for the data corruption of each element in the dataset. However, as it does not take into account the uncertainty in decision-making caused by the simultaneous presence of diverse corrupted data, this work still encounters difficulties in learning robust agents, limiting its applications in real-world scenarios.

In this paper, we propose to use offline data as the observations, thus leveraging their correlations to capture the uncertainty induced by all corrupted data. Considering that (1) diverse corruptions may introduce uncertainties into all elements in the offline dataset, and (2) each element is correlated with the action values (see dashed lines in Figure 1), there is high uncertainty in approximating the action-value function by using various corrupted data. To address this high uncertainty, we propose to leverage all elements in the dataset as observations, based on the graphical model in Figure 1. By using the high correlations between these observations and the action values [27], we can accurately identify the uncertainty of the action-value function.

Motivated by this idea, we propose a robus**t** va**r**iational B**a**yesian inferen**c**e for offlin**e R**L (TRACER) to capture the uncertainty via offline data against all types of data corruptions. Specifically, TRACER first models all data corruptions as uncertainty in the action-value function. Then, to capture such uncertainty, it introduces variational Bayesian inference [28], which uses all offline data as observations to approximate the posterior distribution of the action-value function. Moreover, the corrupted observed data often induce higher uncertainty than clean data, resulting in higher entropy in the distribution of action-value function. Thus, TRACER can use the entropy as an uncertainty measure to effectively distinguish corrupted data from clean data. Based on the entropy-based uncertainty measure, it can regulate the loss associated with corrupted data in approximating the action-value distribution. This approach effectively reduces the influence of corrupted samples, enhancing robustness and performance in clean environments.

This study introduces Bayesian inference into offline RL for data corruptions. It significantly captures the uncertainty caused by diverse corrupted data, thereby improving both robustness and performance in offline RL. Moreover, it is important to note that, unlike traditional Bayesian online and offline RL methods that only model uncertainty from rewards and dynamics [29–35], our approach identifies the

uncertainty of the action-value function regarding states, actions, rewards, and dynamics under data corruptions. We summarize our contributions as follows.

- To the best of our knowledge, this study introduces Bayesian inference into corruption-robust offline RL for the first time. By leveraging all offline data as observations, it can capture uncertainty in the action-value function caused by diverse corrupted data.

- By introducing an entropy-based uncertainty measure, TRACER can distinguish corrupted from clean data, thereby regulating the loss associated with corrupted samples to reduce its influence for robustness.

- Experiment results show that TRACER significantly outperforms several state-of-the-art offline RL methods across a range of both individual and simultaneous data corruptions.

## 2  Preliminaries

**Bayesian RL.** We consider a Markov decision process (MDP), denoted by a tuple $\mathcal{M} = (\mathcal{S}, \mathcal{A}, \mathcal{R}, P, P_0, \gamma)$, where $\mathcal{S}$ is the state space, $\mathcal{A}$ is the action space, $\mathcal{R}$ is the reward space, $P(\cdot|s,a) \in \mathcal{P}(\mathcal{S})$ is the transition probability distribution over next states conditioned on a state-action pair $(s,a)$, $P_0(\cdot) \in \mathcal{P}(\mathcal{S})$ is the probability distribution of initial states, and $\gamma \in [0,1)$ is the discount factor. Note that $\mathcal{P}(\mathcal{S})$ and $\mathcal{P}(\mathcal{A})$ denote the sets of probability distributions on subsets of $\mathcal{S}$ and $\mathcal{A}$, respectively. For simplicity, throughout the paper, we use uppercase letters to refer to random variables and lowercase letters to denote values taken by the random variables. Specifically, $R(s,a)$ denotes the random variable of one-step reward following the distribution $\rho(r|s,a)$, and $r(s,a)$ represents a value of this random variable. We assume that the random variable of one-step rewards and their expectations are bounded by $R_{\max}$ and $r_{\max}$ for any $(s,a) \in \mathcal{S} \times \mathcal{A}$, respectively.

Our goal is to learn a policy that maximizes the expected discounted cumulative return:

$$\pi^* = \arg\max_{\pi \in \mathcal{P}(\mathcal{A})} \mathbb{E}_{s_0 \sim P_0, a_t \sim \pi(\cdot|s_t), R \sim \rho(\cdot|s_t, a_t), s_{t+1} \sim P(\cdot|s_t, a_t)} \left[ \sum_{t=0}^{\infty} \gamma^t R(s_t, a_t) \right].$$

Based on the return, we can define the value function as $V^\pi(s) = \mathbb{E}_{\pi, \rho, P}\left[ \sum_{t=0}^{\infty} \gamma^t R(s_t, a_t) | s_0 = s \right]$, the action-value function as $Q^\pi(s,a) = \mathbb{E}_{R \sim \rho(\cdot|s,a), s' \sim P(\cdot|s,a)}\left[ R(s,a) + \gamma V^\pi(s') \right]$, and the *action-value distribution* [36] as

$$D^\pi(s,a) = \sum_{t=0}^{\infty} \gamma^t R(s_t, a_t | s_0 = s, a_0 = a), \quad \text{with } s_{t+1} \sim P(\cdot|s_t, a_t), a_{t+1} \sim \pi(\cdot|s_{t+1}). \quad (1)$$

Note that $V^\pi(s) = \mathbb{E}_{a \sim \pi}\left[ Q^\pi(s,a) \right] = \mathbb{E}_{a \sim \pi, \rho}\left[ D^\pi(s,a) \right]$.

**Variational Inference.** Variational inference is a powerful method for approximating complex posterior distributions, which is effective for RL to handle the parameter uncertainty and deal with modelling errors [36]. Given an observation $X$ and latent variables $Z$, Bayesian inference aims to compute the posterior distribution $p(Z|X)$. Direct computation of this posterior is often intractable due to the high-dimensional integrals involved. To approximate the true posterior, Bayesian inference introduces a parameterized distribution $q(Z; \phi)$ and minimizes the Kullback-Leibler (KL) divergence $\mathcal{D}_{\mathrm{KL}}(q(Z; \phi) \| p(Z|X))$. Note that minimizing the KL divergence is equivalent to maximizing the evidence lower bound (ELBO) [37, 38]: $\mathrm{ELBO}(\phi) = \mathbb{E}_{q(\mathbf{Z}; \phi)}[\log p(\mathbf{X}, \mathbf{Z}) - \log q(\mathbf{Z}; \phi)]$.

**Offline RL under Diverse Data Corruptions.** In the real world, the data collected by sensors or humans may be subject to diverse corruption due to sensor failures or malicious attacks. Let $b$ and $\mathcal{B}$ denotes the uncorrupted and corrupted dataset with samples $\{(s_t^i, a_t^i, r_t^i, s_{t+1}^i)\}_{i=1}^{N}$, respectively. Each data in $\mathcal{B}$ may be corrupted. We assume that an uncorrupted state follows a state distribution $p_b(\cdot)$, a corrupted state follows $p_{\mathcal{B}}(\cdot)$, an uncorrupted action follows a behavior policy $\pi_b(\cdot|s_t^i)$, a corrupted action is sampled from $\pi_{\mathcal{B}}(\cdot|s_t^i)$, a corrupted reward is sampled from $\rho_{\mathcal{B}}(\cdot|s_t^i, a_t^i)$, and a corrupted next state is drawn from $P_{\mathcal{B}}(\cdot|s_t^i, a_t^i)$. We also denote the uncorrupted and corrupted empirical state-action distributions as $p_b(s_t^i, a_t^i)$ and $p_{\mathcal{B}}(s_t^i, a_t^i)$, respectively. Moreover, we introduce the notations [18, 39] as follows.

$$\tilde{\mathcal{T}}Q(s,a) = \tilde{r}(s,a) + \mathbb{E}_{s' \sim P_{\mathcal{B}}(\cdot|s,a)}\left[ V(s') \right], \quad \tilde{r}(s,a) = \mathbb{E}_{r \sim \rho_{\mathcal{B}}(\cdot|s,a)}[r], \quad (2)$$

$$\tilde{\mathcal{T}}D(s,a) \overset{D}{:=} R(s,a) + \gamma D(s', a'), \quad s' \sim P_{\mathcal{B}}(\cdot|s,a), \ a' \sim \pi_{\mathcal{B}}(\cdot|s), \quad (3)$$

for any $(s, a) \in \mathcal{S} \times \mathcal{A}$ and $Q : \mathcal{S} \times \mathcal{A} \mapsto [0, r_{\max}/(1 - \gamma)]$, where $X \overset{D}{:=} Y$ denotes equality of probability laws, that is the random variable $X$ is distributed according to the same law as $Y$.

To address the diverse data corruptions, based on IQL [26], RIQL [18] introduces quantile estimators with an ensemble of action-value functions $\{Q_{\theta_i}(s, a)\}_{i=1}^K$ and employs a Huber regression [40]:

$$\mathcal{L}_Q(\theta_i) = \mathbb{E}_{(s,a,r,s') \sim \mathcal{B}} \left[ l_H^\kappa \left( r + \gamma V_\psi(s') - Q_{\theta_i}(s, a) \right) \right], \quad l_H^\kappa(x) = \begin{cases} \frac{1}{2\kappa} x^2, & \text{if } |x| \le \kappa \\ |x| - \frac{1}{2}\kappa, & \text{if } |x| > \kappa \end{cases}, \quad (4)$$

$$\mathcal{L}_V(\psi) = \mathbb{E}_{(s,a) \sim \mathcal{B}} \left[ \mathcal{L}_2^\nu \left( Q_\alpha(s, a) - V_\psi(s) \right) \right], \quad \mathcal{L}_2^\nu(x) = |\nu - \mathbb{I}(x < 0)| \cdot x^2. \quad (5)$$

Note that $l_H^\kappa$ is the Huber loss, and $Q_\alpha$ is the $\alpha$-quantile value among $\{Q_{\theta_i}(s, a)\}_{i=1}^K$. RIQL then follows IQL [26] to learn the policy using weighted imitation learning with a hyperparameter $\beta$:

$$\mathcal{L}_\pi(\phi) = \mathbb{E}_{(s,a) \sim \mathcal{B}} \left[ \exp(\beta \cdot A_\alpha(s, a)) \log \pi_\phi(a|s) \right], \quad A_\alpha(s, a) = Q_\alpha(s, a) - V_\psi(s). \quad (6)$$

## 3 Algorithm

We first introduce the Bayesian inference for capturing the uncertainty caused by diverse corrupted data in Section 3.1. Then, we provide our algorithm TRACER with the entropy-based uncertainty measure in Section 3.2. Moreover, we provide the theoretical analysis for robustness, the architecture, and the detailed implementation of TRACER in Appendices A.1, B.1, and B.2, respectively.

### 3.1 Variational Inference for Uncertainty induced by Corrupted Data

We focus on corruption-robust offline RL to learn an agent under diverse data corruptions, i.e., random or adversarial attacks on four elements of the dataset. We propose to use all elements as observations, leveraging the data correlations to simultaneously address the uncertainty. By introducing Bayesian inference framework, our aim is to approximate the posterior distribution of the action-value function.

At the beginning, based on the relationships between the action values and the four elements (i.e., states, actions, rewards, next states) in the offline dataset as shown in Figure 1, we define $D_\theta = D_\theta(S, A, R) \sim p_\theta(\cdot | S, A, R)$, parameterized by $\theta$. Building on the action-value distribution, we can explore how to estimate the posterior of $D_\theta$ using the elements available in the offline data.

Firstly, we start from the actions $\{a_t^i\}_{i=1}^N$ following the corrupted distribution $\pi_\mathcal{B}$ and use them as observations to approximate the posterior of the action-value distribution under a variational inference. As the actions are correlated with the action values and all other elements in the dataset, the likelihood is $p_{\varphi_a}(A|D, S, R, S')$, parameterized by $\varphi_a$. Then, under the variational inference framework, we maximize the posterior and derive to minimize the loss function based on ELBO:

$$\mathcal{L}_{D|A}(\theta, \varphi_a) = \mathbb{E}_{\mathcal{B}, p_\theta} \left[ \mathcal{D}_{\mathrm{KL}} \left( p_{\varphi_a}(A|D_\theta, S, R, S') \, \| \, \pi_\mathcal{B}(A|S) \right) - \mathbb{E}_{A \sim p_{\varphi_a}} \left[ \log p_\theta(D_\theta|S, A, R) \right] \right], \tag{7}$$

where $S$, $R$, and $S'$ follow the offline data distributions $p_\mathcal{B}$, $\rho_\mathcal{B}$, and $P_\mathcal{B}$, respectively.

Secondly, we apply the rewards $\{r_t^i\}_{i=1}^N$ drawn from the corrupted reward distribution $\rho_\mathcal{B}$ as the observations. Considering that the rewards are related to the states, actions, and action values, we model the likelihood as $p_{\varphi_r}(R|D, S, A)$, parameterized by $\varphi_r$. Therefore, we can derive a loss function by following Equation (7):

$$\mathcal{L}_{D|R}(\theta, \varphi_r) = \mathbb{E}_{\mathcal{B}, p_\theta} \left[ \mathcal{D}_{\mathrm{KL}} \left( p_{\varphi_r}(R|D_\theta, S, A) \, \| \, \rho_\mathcal{B}(R|S, A) \right) - \mathbb{E}_{R \sim p_{\varphi_r}} \left[ \log p_\theta(D_\theta|S, A, R) \right] \right], \tag{8}$$

where $S$ and $A$ follow the offline data distributions $p_\mathcal{B}$ and $\pi_\mathcal{B}$, respectively.

Finally, we employ the state $\{s_t^i\}_{i=1}^N$ in the offline dataset following the corrupted distribution $p_\mathcal{B}$ as the observations. Due to the relation of the states, we can model the likelihood as $p_{\varphi_s}(S|D, A, R)$, parameterized by $\varphi_r$. We then have the loss function:

$$\mathcal{L}_{D|S}(\theta, \varphi_s) = \mathbb{E}_{\mathcal{B}, p_\theta} \left[ \mathcal{D}_{\mathrm{KL}} \left( p_{\varphi_s}(S|D_\theta, A, R) \, \| \, p_\mathcal{B}(S) \right) - \mathbb{E}_{S \sim p_{\varphi_s}} \left[ \log p_\theta(D_\theta|S, A, R) \right] \right], \tag{9}$$

where $A$ and $R$ follow the offline data distributions $\pi_\mathcal{B}$ and $\rho_\mathcal{B}$, respectively. We present the detailed derivation process in Appendix A.2.

The goal of first terms in Equations (7), (8), and (9) is to estimate $\pi_\mathcal{B}(A|S)$, $\rho_\mathcal{B}(R|S, A)$, and $p_\mathcal{B}(S)$ using $p_{\varphi_a}(A|D_\theta, S, R, S')$, $p_{\varphi_r}(R|D_\theta, S, A)$, and $p_{\varphi_s}(S|D_\theta, A, R)$, respectively. As we do not

have the explicit expression of distributions $\pi_{\mathcal{B}}$, $\rho_{\mathcal{B}}$, and $p_{\mathcal{B}}$, we cannot directly compute the KL divergence in these first terms. To address this issue, based on the generalized Bayesian inference [41], we can exchange two distributions in the KL divergence. Then, we model all the aforementioned distributions as Gaussian distributions, and use the mean $\mu_\varphi$ and standard deviation $\Sigma_\varphi$ to represent the corresponding $p_\varphi$. For implementation, we directly employ MLPs to output each $(\mu_\varphi, \Sigma_\varphi)$ using the corresponding conditions of $p_\varphi$. Then, based on the KL divergence between two Gaussian distributions, we can derive the loss function as follows.

$$
\mathcal{L}_{\text{first}}(\theta, \varphi_s, \varphi_a, \varphi_r) = \frac{1}{2}\mathbb{E}_{(s,a,r)\sim\mathcal{B}, D_\theta \sim p_\theta} \left[ (\mu_{\varphi_a} - a)^T \Sigma_{\varphi_a}^{-1}(\mu_{\varphi_a} - a) + (\mu_{\varphi_r} - r)^T \Sigma_{\varphi_r}^{-1}(\mu_{\varphi_r} - r) \right.
$$
$$
\left. + (\mu_{\varphi_s} - s)^T \Sigma_{\varphi_s}^{-1}(\mu_{\varphi_s} - s) + \log |\Sigma_{\varphi_a}| \cdot |\Sigma_{\varphi_r}| \cdot |\Sigma_{\varphi_s}| \right], \tag{10}
$$

Moreover, the goal of second terms in Equations (7), (8), and (9) is to maximize the likelihoods of $D_\theta$ given samples $\hat{s} \sim p_{\varphi_s}$, $\hat{a} \sim p_{\varphi_a}$, or $\hat{r} \sim p_{\varphi_r}$. Thus, with $(s, a, r) \sim \mathcal{B}$, we propose minimizing the distance between $D_\theta(\hat{s}, a, r)$ and $D(s, a, r)$, $D_\theta(s, \hat{a}, r)$ and $D(s, a, r)$, and $D_\theta(s, a, \hat{r})$ and $D(s, a, r)$, where $\hat{s} \sim p_{\varphi_s}$, $\hat{a} \sim p_{\varphi_a}$, and $\hat{r} \sim p_{\varphi_r}$. Then, based on [41], we can derive the following loss with any metric $\ell$ to maximize the log probabilities:

$$
\mathcal{L}_{\text{second}}(\theta, \varphi_s, \varphi_a, \varphi_r) = \mathbb{E}_{(s,a,r)\sim\mathcal{B}, \hat{s}\sim p_{\varphi_s}, \hat{a}\sim p_{\varphi_a}, \hat{r}\sim p_{\varphi_r}, D\sim p} \left[ \ell\big(D(s,a,r), D_\theta(s,\hat{a},r)\big) \right.
$$
$$
\left. + \ell\big(D(s,a,r), D_\theta(s,a,\hat{r})\big) + \ell\big(D(s,a,r), D_\theta(\hat{s},a,r)\big) \right]. \tag{11}
$$

### 3.2 Corruption-Robust Algorithm with the Entropy-based Uncertainty Measure

We focus on developing tractable loss functions for implementation in this subsection.

***Learning the Action-Value Distribution based on Temporal Difference (TD).*** Based on [42, 43], we introduce the quantile regression [44] to approximate the action-value distribution in the offline dataset $\mathcal{B}$ using an ensemble model $\{D_{\theta_i}\}_{i=1}^K$. We use Equation (4) to derive the loss as:

$$
\mathcal{L}_D(\theta_i) = \mathbb{E}_{(s,a,r,s')\sim\mathcal{B}} \left[ \frac{1}{N'} \sum_{n=1}^N \sum_{m=1}^{N'} \rho_\tau^\kappa \left( \delta_{\theta_i}^{\tau_n, \tau'_m} \right) \right], \quad \delta_{\theta_i}^{\tau, \tau'} = r + \gamma Z^{\tau'}(s') - D_{\theta_i}^\tau(s,a,r),
$$
$$
\tag{12}
$$

where $\rho_\tau^\kappa(\delta) = |\tau - \mathbb{I}\{\delta < 0\}| \cdot l_H^\kappa(\delta)$ with the threshold $\kappa$, $Z$ denotes the value distribution, $\delta_{\theta_i}^{\tau, \tau'}$ is the sampled TD error based on the parameters $\theta_i$, $\tau$ and $\tau'$ are two samples drawn from a uniform distribution $U([0, 1])$, $D_\theta^\tau(s, a, r) := F_{D_\theta(s,a,r)}^{-1}(\tau)$ is the sample drawn from $p_\theta(\cdot|s, a, r)$, $Z^\tau(s) := F_{Z(s)}^{-1}(\tau)$ is sampled from $p(\cdot|s)$, $F_X^{-1}(\tau)$ is the inverse cumulative distribution function (also known as quantile function) [45] at $\tau$ for the random variable $X$, and $N$ and $N'$ represent the respective number of iid samples $\tau$ and $\tau'$. Notably, based on [43], we have $Q_{\theta_i}(s, a) = \sum_{n=1}^N D_{\theta_i}^{\tau_n}(s, a, r)$.

In addition, if we learn the value distribution $Z$, the action-value distribution can extract the information from the next states based on Equation (12), which is effective for capturing the uncertainty. On the contrary, if we directly use the next states in the offline dataset as the observations, in practice, the parameterized model of the action-value distribution needs to take $(s, a, r, s', a', r', s'')$ as the input data. Thus, the model can compute the action values and values for the sampled TD error in Equation (12). To avoid the changes in the input data caused by directly using next states as observations in Bayesian inference, we draw inspiration from IQL and RIQL to learn a parameterized value distribution. Based on Equations (5) and (12), we derive a new objective as:

$$
\mathcal{L}_Z(\psi) = \mathbb{E}_{(s,a)\sim\mathcal{B}} \left[ \sum_{n=1}^N \mathcal{L}_2^\nu \left( D_\alpha^{\tau_n}(s,a,r) - Z_\psi^{\tau_n}(s) \right) \right], \tag{13}
$$

where $D_\alpha^\tau$ is the $\alpha$-quantile value among $\{D_{\theta_i}^\tau(s,a)\}_{i=1}^K$, and $V_\psi(s) = \sum_{n=1}^N Z_\psi^{\tau_n}(s)$. More details are shown in Appendix B.2. Furthermore, we provide the theoretical analysis in Appendix A.1 to give a value bound between the value distributions under clean and corrupted data.

***Updating the Action-Value Distribution based on Variational Inference for Robustness.*** We discuss the detailed implementation of Equations (10) and (11) based on Equations (12) and (13). As the data

corruptions may introduce heavy-tailed targets [18], we apply the Huber loss to replace all quadratic loss in Equation (10) and the metric $\ell$ in Equation (11), mitigating the issue caused by heavy-tailed targets [46] for robustness. We rewrite Equation (11) as follows.

$$\mathcal{L}_{\text{second}}(\theta_i, \varphi_s, \varphi_a, \varphi_r) = \mathbb{E}_{(s,a,r) \sim \mathcal{B}, \hat{s} \sim p_{\varphi_s}, \hat{a} \sim p_{\varphi_a}, \hat{r} \sim p_{\varphi_r}, D^\tau \sim p} \left[ \sum_{n=1}^N l_H^\kappa \big( D^{\tau_n}(s,a,r), D_{\theta_i}^{\tau_n}(s,\hat{a},r) \big) \right.$$

$$\left. + l_H^\kappa \big( D^{\tau_n}(s,a,r), D_{\theta_i}^{\tau_n}(s,a,\hat{r}) \big) + l_H^\kappa \big( D^{\tau_n}(s,a,r), D_{\theta_i}^{\tau_n}(\hat{s},a,r) \big) \right]. \quad (14)$$

Thus, we have the whole loss function $\mathcal{L}_{D|S,A,R} = \mathcal{L}_{\text{first}}(\theta_i, \varphi_s, \varphi_a, \varphi_r) + \mathcal{L}_{\text{second}}(\theta_i, \varphi_s, \varphi_a, \varphi_r)$ in the generalized variational inference framework. Moreover, based on the assumption of heavy-tailed noise in [18], we have a upper bound of action-value distribution by using the Huber regression loss.

***Entropy-based Uncertainty Measure for Regulating the Loss associated with Corrupted Data.*** To further address the challenge posed by diverse data corruptions, we consider the problem: how to exploit uncertainty to further enhance robustness?

Considering that our goal is to improve performance in clean environments, we propose to reduce the influence of corrupted data, focusing on using clean data to learn agents. Therefore, we provide a two-step plan: (1) distinguishing corrupted data from clean data; (2) regulating the loss associated with corrupted data to reduce its influence, thus enhancing the performance in clean environments.

For (1), as the Shannon entropy for the measures of aleatoric and epistemic uncertainties provides important insight [47–49], and the corrupted data often results in higher uncertainty and entropy of the action-value distribution than the clean data, we use entropy [50] to quantify uncertainties of corrupted and clean data. Furthermore, by considering that the exponential function can amplify the numerical difference in entropy between corrupted and clean data, we propose the use of exponential entropy [51]—a metric of extent of a distribution—to design our uncertainty measure.

Specifically, based on Equation 12, we can use the quantile points $\{\tau_n\}_{n=1}^N$ to learn the corresponding quantile function values $\{D^{\tau_n}\}_{n=1}^N$ drawn from the action-value distribution $p_\theta$. We sort the quantile points and their corresponding function values in ascending order based on the values. Thus, we have the sorted sets $\{\varsigma_n\}_{n=1}^N$, $\{D^{\varsigma_n}\}_{n=1}^N$, and the estimated PDF values $\{\bar{\varsigma}_n\}_{n=1}^N$, where $\bar{\varsigma}_1 = \varsigma_1$ and $\bar{\varsigma}_n = \varsigma_n - \varsigma_{n-1}$ for $1 < n \le N$. Then, we can further estimate differential entropy following [52] (see Appendix A.3 for a detailed derivation).

$$\mathcal{H}(p_{\theta_i}(\cdot|s,a,r)) = - \sum_{n=1}^N \hat{\varsigma}_n \cdot \overline{D}_{\theta_i}^{\varsigma_n}(s,a,r) \cdot \log \hat{\varsigma}_n, \quad (15)$$

where $\hat{\varsigma}_n$ denotes $(\bar{\varsigma}_{n-1} + \bar{\varsigma}_n)/2$ for $1 < n \le N$, and $\overline{D}^{\varsigma_n}$ denotes $D^{\varsigma_n} - D^{\varsigma_{n-1}}$ for $1 < n \le N$.

For (2), TRACER employs the reciprocal value of exponential entropy $1/\exp(\mathcal{H}(p_{\theta_i}))$ to weight the corresponding loss of $\theta_i$ in our proposed whole loss function $\mathcal{L}_{D|S,A,R}$. Therefore, during the learning process, TRACER can regulate the loss associated with corrupted data and focus on minimizing the loss associated with clean data, enhancing robustness and performance in clean environments. Note that we normalize entropy values by dividing the mean of samples (i.e., quantile function values) drawn from action-value distributions for each batch. In Figure 3, we show the relationship of entropy values of corrupted and clean data estimated by Equation (15) during the learning process. The results illustrate the effectiveness of the entropy-weighted technique for data corruptions.

***Updating the Policy based on the Action-Value Distribution.*** We directly applies the weighted imitation learning technique in Equation (6) to learn the policy. As $Q_\alpha(s,a)$ is the $\alpha$-quantile value among $\{Q_{\theta_i}(s,a)\}_{i=1}^K = \left\{ \sum_{n=1}^N D_{\theta_i}^{\tau_n}(s,a,r) \right\}_{i=1}^K$ and $V_\psi(s) = \sum_{n=1}^N Z_\psi^{\tau_n}(s)$, we have

$$\mathcal{L}_\pi(\phi) = \mathbb{E}_{(s,a) \sim \mathcal{B}} \left[ \exp\left( \beta \cdot Q_\alpha(s,a) - \beta \cdot \sum_{n=1}^N Z_\psi^{\tau_n}(s) \right) \cdot \log \pi_\phi(a|s) \right]. \quad (16)$$

Table 1: Average scores and standard errors under random and adversarial simultaneous corruptions.

| Env | Corrupt | BC | EDAC | MSG | UWMSG | CQL | IQL | RIQL | TRACER (ours) |
|---|---|---|---|---|---|---|---|---|---|
| Halfcheetah | random | 23.17 ± 0.43 | 1.70 ± 0.80 | 9.97 ± 3.44 | 8.31 ± 1.25 | 14.25 ± 1.39 | 24.82 ± 0.57 | 29.94 ± 1.00 | **33.04 ± 0.42** |
| | advers | 16.37 ± 0.32 | 0.90 ± 0.30 | 3.60 ± 0.89 | 3.13 ± 0.85 | 5.61 ± 2.21 | 11.06 ± 0.45 | 17.85 ± 1.39 | **19.72 ± 2.80** |
| Walker2d | random | 13.77 ± 1.05 | −0.13 ± 0.01 | −0.15 ± 0.11 | 4.36 ± 1.95 | 0.63 ± 0.36 | 12.35 ± 2.03 | 17.42 ± 2.95 | **23.62 ± 2.33** |
| | advers | 6.75 ± 0.33 | −0.17 ± 0.01 | 3.77 ± 1.09 | 4.19 ± 2.82 | 4.23 ± 1.35 | 16.61 ± 2.73 | 9.20 ± 1.40 | **17.21 ± 1.62** |
| Hopper | random | 18.49 ± 0.52 | 0.80 ± 0.01 | 15.84 ± 2.47 | 12.22 ± 2.11 | 3.16 ± 1.07 | 25.28 ± 15.34 | 22.50 ± 10.01 | **28.83 ± 7.06** |
| | advers | 17.34 ± 1.00 | 0.80 ± 0.01 | 12.14 ± 0.71 | 10.43 ± 0.94 | 0.10 ± 0.34 | 19.56 ± 1.08 | 24.71 ± 6.20 | **24.80 ± 7.14** |
| Average score | | 15.98 | 0.65 | 7.53 | 7.11 | 4.66 | 18.28 | 20.27 | **24.54** |

Table 2: Average score under diverse random corruptions.

| Env | Corrupted Element | BC | EDAC | MSG | UWMSG | CQL | IQL | RIQL | TRACER (ours) |
|---|---|---|---|---|---|---|---|---|---|
| Halfcheetah | observation | 33.4 ± 1.8 | 2.1 ± 0.5 | −0.2 ± 2.2 | 2.9 ± 0.1 | 9.0 ± 7.5 | 21.4 ± 1.9 | 27.3 ± 2.4 | **34.2 ± 0.9** |
| | action | 36.2 ± 0.3 | 47.4 ± 1.3 | 52.0 ± 0.9 | **56.0 ± 0.4** | 19.9 ± 21.3 | 42.2 ± 1.9 | 42.9 ± 0.6 | 42.9 ± 0.6 |
| | reward | 35.8 ± 0.9 | 38.6 ± 0.3 | 17.5 ± 16.4 | 35.6 ± 0.4 | 32.6 ± 19.6 | 42.3 ± 0.4 | **43.6 ± 0.6** | 40.0 ± 1.1 |
| | dynamics | 35.8 ± 0.9 | 1.5 ± 0.2 | 1.7 ± 0.4 | 2.9 ± 0.1 | 29.2 ± 4.0 | 36.7 ± 1.8 | 43.1 ± 0.2 | **43.8 ± 3.0** |
| Walker2d | observation | 9.6 ± 3.9 | −0.2 ± 0.3 | −0.4 ± 0.1 | 6.2 ± 0.5 | 19.4 ± 1.6 | 27.2 ± 5.1 | 28.4 ± 7.7 | **32.7 ± 2.8** |
| | action | 18.1 ± 2.1 | 83.2 ± 1.9 | 25.3 ± 10.6 | 31.5 ± 10.6 | 62.7 ± 7.2 | 71.3 ± 7.8 | 84.6 ± 3.3 | **86.7 ± 6.2** |
| | reward | 16.0 ± 7.4 | 4.3 ± 3.6 | 18.4 ± 9.5 | 62.0 ± 3.7 | 69.4 ± 7.4 | 65.3 ± 8.4 | 83.2 ± 2.6 | **85.5 ± 3.1** |
| | dynamics | 16.0 ± 7.4 | −0.1 ± 0.0 | 7.4 ± 3.7 | 0.2 ± 0.0 | −0.2 ± 0.1 | 17.7 ± 7.3 | **78.2 ± 1.8** | 75.9 ± 1.8 |
| Hopper | observation | 21.5 ± 2.9 | 1.0 ± 0.5 | 6.9 ± 5.0 | 12.8 ± 0.4 | 42.8 ± 7.0 | 52.0 ± 16.6 | 62.4 ± 1.8 | **62.7 ± 8.2** |
| | action | 22.8 ± 7.0 | **100.8 ± 0.5** | 37.6 ± 6.5 | 53.4 ± 5.4 | 69.8 ± 4.5 | 76.3 ± 15.4 | 90.6 ± 5.6 | 92.8 ± 2.5 |
| | reward | 19.5 ± 3.4 | 2.6 ± 0.7 | 24.9 ± 4.3 | 60.8 ± 7.5 | 70.8 ± 8.9 | 69.7 ± 18.8 | 84.8 ± 13.1 | **85.7 ± 1.4** |
| | dynamics | 19.5 ± 3.4 | 0.8 ± 0.0 | 12.4 ± 4.9 | 6.1 ± 1.3 | 0.8 ± 0.0 | 1.3 ± 0.5 | **51.5 ± 8.1** | 49.8 ± 5.3 |
| Average score | | 23.7 | 23.5 | 17.0 | 27.5 | 35.5 | 43.6 | 60.0 | **61.1** |

Table 3: Average score under diverse adversarial corruptions.

| Env | Corrupted Element | BC | EDAC | MSG | UWMSG | CQL | IQL | RIQL | TRACER (ours) |
|---|---|---|---|---|---|---|---|---|---|
| Halfcheetah | observation | 34.5 ± 1.5 | 1.1 ± 0.3 | 1.1 ± 0.2 | 1.9 ± 0.1 | 5.0 ± 11.6 | 32.6 ± 2.7 | 35.7 ± 4.2 | **36.8 ± 2.1** |
| | action | 14.0 ± 1.1 | 32.7 ± 0.7 | **37.3 ± 0.7** | 36.2 ± 1.0 | −2.3 ± 1.2 | 27.5 ± 0.3 | 31.7 ± 1.7 | 33.4 ± 1.2 |
| | reward | 35.8 ± 0.9 | 40.3 ± 0.5 | **47.7 ± 0.4** | 43.8 ± 0.3 | −1.7 ± 0.3 | 42.6 ± 0.4 | 44.1 ± 0.8 | 41.9 ± 0.2 |
| | dynamics | 35.8 ± 0.9 | −1.3 ± 0.1 | −1.5 ± 0.0 | 5.0 ± 2.2 | −1.6 ± 0.0 | 26.7 ± 0.7 | 35.8 ± 2.1 | **36.2 ± 1.2** |
| Walker2d | observation | 12.7 ± 5.9 | −0.0 ± 0.1 | 2.9 ± 2.7 | 6.3 ± 0.7 | 61.8 ± 7.4 | 37.7 ± 13.0 | **70.0 ± 5.3** | **70.0 ± 6.7** |
| | action | 5.4 ± 0.4 | 41.9 ± 24.0 | 5.4 ± 0.9 | 5.9 ± 0.4 | 27.0 ± 7.5 | 27.5 ± 0.6 | 66.1 ± 4.6 | **69.3 ± 4.6** |
| | reward | 16.0 ± 7.4 | 57.3 ± 33.2 | 9.6 ± 4.9 | 35.1 ± 10.5 | 67.0 ± 6.1 | 73.5 ± 4.9 | 85.0 ± 1.5 | **88.9 ± 4.7** |
| | dynamics | 16.0 ± 7.4 | 4.3 ± 0.9 | 0.1 ± 0.2 | 1.8 ± 0.2 | 3.9 ± 1.4 | −0.1 ± 0.1 | 60.6 ± 21.8 | **64.0 ± 16.5** |
| Hopper | observation | 21.6 ± 7.1 | 36.2 ± 16.2 | 16.0 ± 2.8 | 15.0 ± 1.3 | **78.0 ± 6.5** | 32.8 ± 6.4 | 50.8 ± 7.6 | 64.5 ± 3.7 |
| | action | 15.5 ± 2.2 | 25.7 ± 3.8 | 23.0 ± 2.1 | 27.7 ± 1.3 | 32.2 ± 7.6 | 37.9 ± 4.8 | 63.6 ± 7.3 | **67.2 ± 3.8** |
| | reward | 19.5 ± 3.4 | 21.2 ± 1.9 | 22.6 ± 2.8 | 30.3 ± 4.2 | 49.6 ± 12.3 | 57.3 ± 9.7 | **65.8 ± 9.8** | 64.3 ± 1.5 |
| | dynamics | 19.5 ± 3.4 | 0.6 ± 0.0 | 0.6 ± 0.0 | 0.7 ± 0.0 | 0.6 ± 0.0 | 1.3 ± 1.1 | **65.7 ± 21.1** | 61.1 ± 6.2 |
| Average score | | 20.5 | 21.7 | 13.7 | 17.5 | 26.6 | 33.1 | 56.2 | **58.1** |

## 4 Experiments

In this section, we show the effectiveness of TRACER across various simulation tasks using diverse corrupted offline datasets. Firstly, we provide our experiment setting, focusing on the corruption settings for offline datasets. Then, we illustrate how TRACER significantly outperforms previous state-of-the-art approaches under a range of both individual and simultaneous data corruptions. Finally, we conduct validation experiments and ablation studies to show the effectiveness of TRACER.

### 4.1 Experiment Setting

Building upon RIQL [18], we use two hyperparameters, i.e., corruption rate $c \in [0, 1]$ and corruption scale $\epsilon$, to control the corruption level. Then, we introduce the *random corruption* and *adversarial corruption* in four elements (i.e., states, actions, rewards, next states) of offline datasets. The implementation of random corruption is to add random noise to elements of a $c$ portion of the offline datasets, and the implementation of adversarial corruption follows the Projected Gradient Descent attack [53, 54] using pretrained value functions. Note that unlike other adversarial corruptions, the adversarial reward corruption multiplies $-\epsilon$ to the clean rewards instead of using gradient optimization. We also introduce the random or adversarial simultaneous corruption, which refers to random or adversarial corruption simultaneously present in four elements of the offline datasets. We apply the corruption rate $c = 0.3$ and corruption scale $\epsilon = 1.0$ in our experiments.

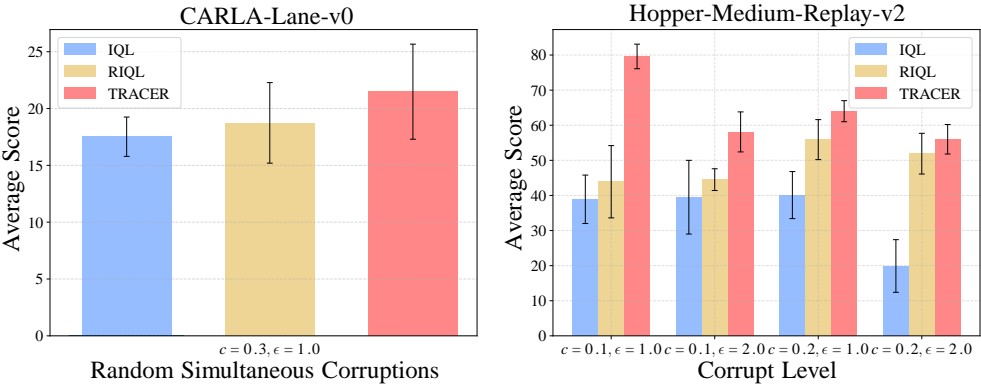

Figure 2: In the left, we report the means and standard deviations on CARLA under random simultaneous corruptions. In the right, we report the results with random simultaneous corruptions against different corruption levels.

We conduct experiments on D4RL benchmark [55]. Referring to RIQL, we train all agents for 3000 epochs on the 'medium-replay-v2' dataset, which closely mirrors real-world applications as it is collected during the training of a SAC agent. Then, we evaluate agents in clean environments, reporting the average normalized performance over four random seeds. See Appendix C for detailed information. The algorithms we compare include: (1) CQL [7] and IQL [26], offline RL algorithms using a twin Q networks. (2) EDAC [56] and MSG [57], offline RL algorithms using ensemble Q networks (number of ensembles $> 2$). (3) UWMSG [17] and RIQL, state-of-the-arts in corruption-robust offline RL. Note that EDAC, MSG, and UWMSG are all uncertainty-based offline RL algorithms.

## 4.2 Main results under Diverse Data Corruptions

We conduct experiments on MuJoCo [58] (see Tables 1, 2, and 3, which highlight the *highest* results) and CARLA [59] (see the left of Figure 2) tasks from D4RL under diverse corruptions to show the superiority of TRACER. In Table 1, we report all results under random or adversarial simultaneous data corruptions. These results show that TRACER significantly outperforms other algorithms in all tasks, achieving an average score improvement of $+\mathbf{21.1}\%$. In the left of Figure 2, results on 'CARLA-lane_v0' under random simultaneous corruptions also illustrate the superiority of TRACER. See Appendix C.2 for details.

**Random Corruptions.** We report the results under random simultaneous data corruptions of all algorithms in Table 1. Such results demonstrate that TRACER achieves an average score gain of $+\mathbf{22.4}\%$ under the setting of random simultaneous corruptions. Based on the results, it is clear that many offline RL algorithms, such as EDAC, MSG, and CQL, suffer the performance degradation under data corruptions. Since UWMSG is designed to defend the corruptions in rewards and dynamics, its performance degrades when faced with the stronger random simultaneous corruption. Moreover, we report results across a range of individual random data corruptions in Table 2, where TRACER outperforms previous algorithms in 7 out of 12 settings. We then explore hyperparameter tuning on Hopper task and further improve TRACER's results, demonstrating its potential for performance gains. We provide details in Appendix C.3.1.

**Adversarial Corruptions.** We construct experiments under adversarial simultaneous corruptions to evaluate the robustness of TRACER. The results in Table 1 show that TRACER surpasses others by a significant margin, achieving an average score improvement of $+\mathbf{19.3}\%$. In these simultaneous corruption, many algorithms experience more severe performance degradation compared to the random simultaneous corruption, which indicates that adversarial attacks are more damaging to the reliability of algorithms than random noise. Despite these challenges, TRACER consistently achieves significant performance gains over other methods. Moreover, we provide the results across a range of individual adversarial data corruptions in Table 3, where TRACER outperforms previous algorithms in 7 out of 12 settings. We also explore hyperparameter tuning on Hopper task and further improve TRACER's results, demonstrating its potential for performance gains. See Appendix C.3.1 for details.

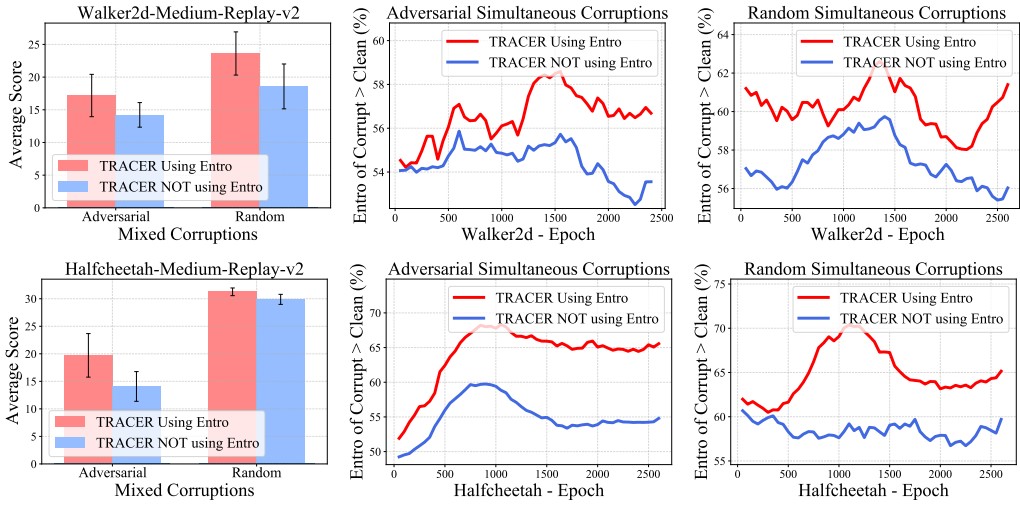

Figure 3: In the first column, we report the mean and standard deviation to show the superiority of using the entropy-based uncertainty measure. In the second and third columns, we report the results over three seeds to show the higher entropy of corrupted data compared to clean data during training.

## 4.3 Evaluation of TRACER under Various Corruption Levels

Building upon RIQL, we further extend our experiments to include Mujoco datasets with various corruption levels, using different corruption rates $c$ and scales $\epsilon$. We report the average scores and standard deviations over four random seeds in the right of Figure 2, using batch sizes of 256.

Results in the right of Figure 2 demonstrate that TRACER significantly outperforms baseline algorithms in **all tasks** under random simultaneous corruptions with various corruption levels. It achieves an average score improvement of $+\mathbf{33.6}\%$. Moreover, as the corruption levels increase, the slight decrease in TRACER's results indicates that while TRACER is robust to simultaneous corruptions, its performance depends on the extent of corrupted data it encounters. We also evaluate TRACER in different scales of corrupted data and provide the results in Appendix C.4.1.

## 4.4 Evaluation of the Entropy-based Uncertainty Measure

We evaluate the entropy-based uncertainty measure in action-value distributions to show: (1) is the uncertainty caused by corrupted data higher than that of clean data? (2) is regulating the loss associated with corrupted data effective in improving performance?

For (1), we first introduce labels indicating whether the data is corrupted. Importantly, these labels are not used by agents during the training process. Then, we estimate entropy values of labelled corrupted and clean data in each batch based on Equation (15). Thus, we can compare entropy values to compute results, showing how many times the entropy of the corrupted data is higher than that of clean data. Specifically, we evaluate the accuracy every 50 epochs over 3000 epochs. For each evaluation, we sample 500 batches to compute the average entropy of corrupted and clean data. Each batch consists of 32 clean and 32 corrupted data. We illustrate the curves over three seeds in the second and third columns of Figure 3, where each point shows how many of the 500 batches have higher entropy for corrupted data than that of clean data.

Figure 3 indicates an oscillating upward trend of TRACER's measurement accuracy using entropy (*TRACER Using Entro*) under simultaneous corruptions, demonstrating that using the entropy-based uncertainty measure can effectively distinguish corrupted data from clean data. These curves also reveal that even in the absence of any constraints on entropy (*TRACER NOT using Entro*), the entropy associated with corrupted data tends to exceed that of clean data.

For (2), in the first column of Figure 3, these results demonstrate that TRACER using the entropy-based uncertainty measure can effectively reduce the influence of corrupted data, thereby enhancing robustness and performance against all corruptions. We provide detailed information for this evaluation in Appendix C.4.2.

## 5    Related Work

**Robust RL.**    Robust RL can be categorized into two types: testing-time robust RL and training-time robust RL. Testing-time robust RL [19, 20] refers to training a policy on clean data and ensuring its robustness by testing in an environment with random noise or adversarial attacks. Training-time robust RL [16, 17] aims to learn a robust policy in the presence of random noise or adversarial attacks during training and evaluate the policy in a clean environment. In this paper, we focus on training-time robust RL under the offline setting, where the offline training data is subject to various data corruptions, also known as *corruption-robust offline RL*.

**Corruption-Robust RL.**    Some theoretical work on corruption-robust online RL [60–63] aims to analyze the sub-optimal bounds of learned policies under data corruptions. However, these studies primarily address simple bandits or tabular MDPs and focus on the reward corruption. Some further work [64, 65] extends the modeling problem to more general MDPs and begins to investigate the corruption in transition dynamics.

It is worth noting that corruption-robust offline RL has not been widely studied. UWMSG [17] designs a value-based uncertainty-weighting technique, thus using the weight to mitigate the impact of corrupted data. RIQL [18] further extends the data corruptions to all four elements in the offline dataset, including states, actions, rewards, and next states (dynamics). It then introduces quantile estimators with an ensemble of action-value functions and employs a Huber regression based on IQL [26], alleviating the performance degradation caused by corrupted data.

**Bayesian RL.**    Bayesian RL integrates the Bayesian inference with RL to create a framework for decision-making under uncertainty [28]. It is important to highlight that Bayesian RL is divided into two categories for different uncertainties: the *parameter uncertainty* in the learning of models [66, 67] and the ***inherent uncertainty*** from the data/environment in the distribution over returns [68, 69]. In this paper, we focus on capturing the latter.

For the latter uncertainty, in model-based Bayesian RL, many approaches [68, 70, 71] explicitly model the transition dynamics and using Bayesian inference to update the model. It is useful when dealing with complex environments for sample efficiency. In model-free Bayesian RL, value-based methods [69, 72] use the reward information to construct the posterior distribution of the action-value function. Besides, policy gradient methods [73, 74] use information of the return to construct the posterior distribution of the policy. They directly apply Bayesian inference to the value function or policy without explicitly modeling transition dynamics.

**Offline Bayesian RL.**    offline Bayesian RL integrates Bayesian inference with offline RL to tackle the challenges of learning robust policies from static datasets without further interactions with the environment. Many approaches [75–77] use Bayesian inference to model the transition dynamics or guide action selection for adaptive policy updates, thereby avoiding overly conservative estimates in the offline setting. Furthermore, recent work [78] applies variational Bayesian inference to learn the model of transition dynamics, mitigating the distribution shift in offline RL.

## 6    Conclusion

In this paper, we investigate and demonstrate the robustness and effectiveness of introducing Bayesian inference into offline RL to address the challenges posed by data corruptions. By leveraging Bayesian techniques, our proposed approach TRACER captures the uncertainty caused by diverse corrupted data. Moreover, the use of entropy-based uncertainty measure in TRACER can distinguish corrupted data from clean data. Thus, TRACER can regulate the loss associated with corrupted data to reduce its influence, improving performance in clean environments. Our extensive experiments demonstrate the potential of Bayesian methods in developing reliable decision-making.

Regarding the limitations of TRACER, although it achieves significant performance improvement under diverse data corruptions, future work could explore more complex and realistic data corruption scenarios and related challenges, such as the noise in the preference data for RLHF and adversarial attacks on safety-critical driving decisions. Moreover, we look forward to the continued development and optimization of uncertainty-based corrupted-robust offline RL, which could further enhance the effectiveness of TRACER and similar approaches for increasingly complex real-world scenarios.

## Acknowledgments

We would like to thank all the anonymous reviewers for their insightful comments. This work was supported in part by National Key R&D Program of China under contract 2022ZD0119801, National Nature Science Foundations of China grants U23A20388 and 62021001, and DiDi GAIA Collaborative Research Funds.

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

## Appendix / supplemental material

## A    Proofs

We provide a proof for the upper bound of value distributions and derivations of loss functions. See Table 4 for all notations.

### A.1    Proof for Value Bound

To prove an upper bound of the value distribution, we introduce an assumption from [18] below.

**Assumption A.1.**  *[18] Let $\zeta = \sum_{i=1}^{N} \left( 2\zeta_i + \log \zeta_i' \right)$ denote the cumulative corruption level, where $\zeta_i$ and $\zeta_i'$ are defined as*

$$\left\| \mathcal{T} V\left(s_i\right) - \tilde{\mathcal{T}} V\left(s_i\right) \right\|_\infty \leq \zeta_i, \quad \max \left\{ \frac{\pi_{\mathcal{B}}\left(a \mid s_i\right)}{\pi_b\left(a \mid s_i\right)}, \frac{\pi_b\left(a \mid s_i\right)}{\pi_{\mathcal{B}}\left(a \mid s_i\right)} \right\} \leq \zeta_i', \quad \forall a \in \mathcal{A}.$$

*Here $\| \cdot \|_\infty$ means taking supremum over $V : \mathcal{S} \mapsto [0, r_{\max}/(1 - \gamma)]$.*

Note that $\{\zeta_i'\}_i = 1^N$ can quantify the corruption level of states and actions, and $\{\zeta_i\}_i = 1^N$ can quantify the corruption level of rewards and next states (transition dynamics).

Then, we provide the following assumption.

**Assumption A.2.**  *There exists an $M > 0$ such that*

$$\max \{ \frac{d^{\pi_E}(s,a)}{p_b(s,a)}, \frac{d^{\tilde{\pi}_E}(s,a)}{p_{\mathcal{B}}(s,a)}, \frac{d^{\pi_E}(s)}{d^{\tilde{\pi}_E}(s)}, \frac{d^{\pi_{IQL}}(s)}{d^{\pi_E}(s)}, \frac{d^{\tilde{\pi}_{IQL}}(s)}{d^{\tilde{\pi}_E}(s)} \} \leq M, \quad \forall (s,a) \in \mathcal{S} \times \mathcal{A},$$

*where $\pi_{\mathrm{E}}(a \mid s) \propto \pi_b(a \mid s) \cdot \exp \left( \beta \cdot [\mathcal{T}Q^* - V^*](s,a) \right)$ is the policy of clean data, $\pi_{IQL}$ in Equation (17) is the policy following IQL's supervised policy learning scheme under clean data, $\tilde{\pi}_{\mathrm{IQL}} = \arg\min_\pi \mathbb{E}_{s \sim \mathcal{B}} [\mathrm{KL} \left( \tilde{\pi}_{\mathrm{E}}(\cdot \mid s), \pi(\cdot \mid s) \right)]$ is the policy under corrupted data, and $\tilde{\pi}_{\mathrm{E}}(a \mid s) \propto \pi_{\mathcal{B}}(a \mid s) \cdot \exp \left( \beta \cdot \left[ \tilde{\mathcal{T}}Q^* - V^* \right](s,a) \right)$ is the policy of corrupted data.*

$$\pi_{\mathrm{IQL}} = \arg\max_\pi \ \mathbb{E}_{(s,a)\sim b} \left[ \exp \left( \beta \cdot [\mathcal{T}Q^* - V^*](s,a) \right) \log \pi(a \mid s) \right] \tag{17}$$

$$= \arg\min_\pi \ \mathbb{E}_{s \sim b} \left[ \mathcal{D}_{\mathrm{KL}} \left( \pi_{\mathrm{E}}(\cdot \mid s), \pi(\cdot \mid s) \right) \right]. \tag{18}$$

As TRACER directly applies the weighted imitation learning technique from IQL to learn the policy, we can use $\tilde{\pi}_{\mathrm{IQL}}$ as the policy learned by TRACER under data corruptions, akin to RIQL. Assumption A.2 requires that each pair, including the policy $\pi_E$ and the clean data $b$, the policy $\tilde{\pi}_E$ and the corrupted dataset $\mathcal{B}$, $\pi_E$ and $\tilde{\pi}_E$, $\pi_E$ and $\pi_{\mathrm{IQL}}$, and $\tilde{\pi}_E$ and $\tilde{\pi}_{\mathrm{IQL}}$, has good coverage. It is similar to the coverage condition in [18]. Based on Assumptions A.1 and A.2, we can derive the following theorem to show the robustness of our approach using the supervised policy learning scheme and learning the action-value distribution.

**Lemma A.3.**  *(Performance Difference) For any $\tilde{\pi}$ and $\pi$, we have*

$$Z^{\tilde{\pi}}(s) - Z^\pi(s) = \frac{1}{1 - \gamma} \mathbb{E}_{(s,a)\sim d^{\tilde{\pi}}, \tilde{\pi}} \left[ D^\pi(s,a,r) - Z^\pi(s) \right]. \tag{19}$$

*Proof.* Based on [79], for any $\tilde{\pi}$ and $\pi$, we have

$$Z^{\tilde{\pi}}(s) = \sum_{t=0}^{\infty} \gamma^t \mathbb{E}_{(S_t, A_t) \sim P, \tilde{\pi}} \left[ R(S_t, A_t) | S_0 = s \right] \tag{20}$$

$$= \sum_{t=0}^{\infty} \gamma^t \mathbb{E}_{(S_t, A_t) \sim P, \tilde{\pi}} \left[ R(S_t, A_t) + Z^{\pi}(S_t) - Z^{\pi}(S_t) | S_0 = s \right] \tag{21}$$

$$= \sum_{t=0}^{\infty} \gamma^t \mathbb{E}_{(S_t, A_t, S_{t+1}) \sim P, \tilde{\pi}} \left[ R(S_t, A_t) + \gamma Z^{\pi}(S_{t+1}) - Z^{\pi}(S_t) | S_0 = s \right] + Z^{\pi}(s) \tag{22}$$

$$= Z^{\pi}(s) + \sum_{t=0}^{\infty} \gamma^t \mathbb{E}_{(S_t, A_t) \sim P, \tilde{\pi}} \left[ D^{\pi}(S_t, A_t, R_t) - Z^{\pi}(S_t) | S_0 = s \right] \tag{23}$$

$$= Z^{\pi}(s) + \frac{1}{1 - \gamma} \mathbb{E}_{(s,a) \sim d^{\tilde{\pi}}, \tilde{\pi}} \left[ D^{\pi}(s, a, r) - Z^{\pi}(s) \right]. \tag{24}$$

$\square$

**Theorem A.4.** *(Robustness). Under Assumptions A.1 and A.2, we have*

$$W_1(q^{\tilde{\pi}}, q^{\pi}) \le \frac{2M r_{\max}}{(1 - \gamma)^2} \left( \sqrt{1 - e^{-\frac{M\varsigma}{N}}} + \sqrt{1 - e^{-\epsilon_1}} + \sqrt{1 - e^{-\epsilon_2}} \right), \tag{25}$$

*where $q^{\pi}$ is the value distribution, $W_1(\cdot, \cdot)$ is the Wasserstein distance [80] for measuring the distribution difference, $\epsilon_1 = \mathbb{E}_{s \sim b} \left[ \mathcal{D}_{KL} \left( \pi_E(\cdot|s) \,\|\, \pi_{IQL}(\cdot|s) \right) \right]$, and $\epsilon_2 = \mathbb{E}_{s \sim \mathcal{B}} \left[ \mathcal{D}_{KL} \left( \tilde{\pi}_E(\cdot|s) \,\|\, \tilde{\pi}_{IQL}(\cdot|s) \right) \right]$.*

*Proof.* The definition of Wasserstein metric is $W_p(P, Q) = \left( \inf_{\gamma \in \Gamma(P,Q)} \int_x \int_y \|x - y\|_p^p d\gamma(x, y) \right)^{\frac{1}{p}}$.

The value distribution $q^{\pi}$, also denoted by $p(\cdot|s)$ in Section 3.2 of the main text, is an expactation of action value distribution $\mathbb{E}_{a \sim \pi, r \sim \rho}[p_\theta(\cdot|s, a, r)]$.

Note that as TRACER directly applies the weighted imitation learning technique from IQL to learn the policy (see Section 3.2), we can use $\tilde{\pi}_{IQL}$ as the policy learned by TRACER under data corruptions, akin to RIQL.

Let $q^{\tilde{\pi}_{IQL}}$ and $q^{\pi_{IQL}}$ be the value distributions of learned policies following IQL's supervised policy learning scheme under corrupted and clean data, respectively. $Z^{\tilde{\pi}_{IQL}} \sim q^{\tilde{\pi}_{IQL}}$ and $Z^{\pi_{IQL}} \sim q^{\pi_{IQL}}$. Let $p^{\pi_E}$ be the value distribution of the policy of clean data, $p^{\tilde{\pi}_E}$ be the value distribution of the policy of corrupted data, $Z^{\pi_E} \sim p^{\pi_E}$, and $Z^{\tilde{\pi}_E} \sim p^{\tilde{\pi}_E}$. We have

$$W_1(q^{\tilde{\pi}_{IQL}}, q^{\pi_{IQL}}) \le W_1(q^{\tilde{\pi}_{IQL}}, p^{\tilde{\pi}_E}) + W_1(p^{\tilde{\pi}_E}, q^{\pi_{IQL}}) \tag{26}$$

$$\le W_1(q^{\tilde{\pi}_{IQL}}, p^{\tilde{\pi}_E}) + W_1(p^{\tilde{\pi}_E}, p^{\pi_E}) + W_1(p^{\pi_E}, q^{\pi_{IQL}}) \tag{27}$$

Moreover, based on the Bretagnolle–Huber inequality, we have $d_{TV}(P,Q) \leq \sqrt{1 - \exp(-\mathcal{D}_{KL}(P\|Q))}$. For $W_1(p^{\tilde{\pi}_E}, p^{\pi_E})$, we have

$$W_1(p^{\tilde{\pi}_E}, p^{\pi_E}) = \inf_{\gamma \in \Gamma(p^{\tilde{\pi}_E}, p^{\pi_E})} \int_{Z^{\tilde{\pi}_E}} \int_{Z^{\pi_E}} ||Z^{\tilde{\pi}_E} - Z^{\pi_E}|| d\gamma(Z^{\tilde{\pi}_E}, Z^{\pi_E}) \tag{28}$$

$$= \inf_{\gamma \in \Gamma(d^{\tilde{\pi}_E}, d^{\pi_E})} \int_s \int_{s'} ||Z^{\tilde{\pi}_E}(s) - Z^{\pi_E}(s')|| d\gamma(s, s') \tag{29}$$

$$\leq M * \int_s |Z^{\tilde{\pi}_E}(s) - Z^{\pi_E}(s)| d^{\tilde{\pi}_E}(s) ds \tag{30}$$

$$= M * \mathbb{E}_{s \sim d^{\tilde{\pi}_E}} |Z^{\tilde{\pi}_E}(s) - Z^{\pi_E}(s)| \tag{31}$$

$$= M * \frac{1}{1-\gamma} \mathbb{E}_{s \sim d^{\tilde{\pi}_E}, a \sim \tilde{\pi}_E(\cdot|s)} [|Z^{\pi_E}(s) - D^{\pi_E}(s, a, r)|] \tag{32}$$

$$\overset{A.3}{=} M * \frac{1}{1-\gamma} \mathbb{E}_{s \sim d^{\tilde{\pi}_E}} [\mathbb{E}_{a \sim \pi_E(\cdot|s)}[D^{\pi_E}(s, a, r)] - \mathbb{E}_{a \sim \tilde{\pi}_E(\cdot|s)}[D^{\pi_E}(s, a, r)]] \tag{33}$$

$$\leq \frac{Mr_{\max}}{(1-\gamma)^2} \mathbb{E}_{s \sim d^{\tilde{\pi}_E}} ||\tilde{\pi}_E(\cdot|s) - \pi_E(\cdot|s)||_1 \tag{34}$$

$$\leq \frac{2Mr_{\max}}{(1-\gamma)^2} \mathbb{E}_{s \sim d^{\tilde{\pi}_E}} \sqrt{1 - e^{-\mathcal{D}_{\mathrm{KL}}(\tilde{\pi}_E(\cdot|s), \pi_E(\cdot|s))}} \tag{35}$$

$$\leq \frac{2Mr_{\max}}{(1-\gamma)^2} \sqrt{\mathbb{E}_{s \sim d^{\tilde{\pi}_E}} (1 - e^{-\mathcal{D}_{\mathrm{KL}}(\tilde{\pi}_E(\cdot|s), \pi_E(\cdot|s))})} \tag{36}$$

$$\leq \frac{2Mr_{\max}}{(1-\gamma)^2} \sqrt{1 - e^{-\mathbb{E}_{s \sim d^{\tilde{\pi}_E}} \mathcal{D}_{\mathrm{KL}}(\tilde{\pi}_E(\cdot|s), \pi_E(\cdot|s))}} \tag{37}$$

$$= \frac{2Mr_{\max}}{(1-\gamma)^2} \sqrt{1 - e^{-\mathbb{E}_{(s,a) \sim d^{\tilde{\pi}_E}} [\log \frac{\tilde{\pi}_E(a|s)}{\pi_E(a|s)}]}}. \tag{38}$$

Moreover, we have

$$\frac{\tilde{\pi}_{\mathrm{E}}(a_i|s_i)}{\pi_{\mathrm{E}}(a_i|s_i)} = \frac{\tilde{\pi}_{\mathcal{B}}(a_i|s_i) \cdot \exp(\beta \cdot [\tilde{\mathcal{T}}V^* - V^*](s,a))}{\pi_{\mathcal{B}}(a_i|s_i) \cdot \exp(\beta \cdot [\tilde{\mathcal{T}}V^* - V^*](s,a))} \tag{39}$$

$$\times \frac{\sum_{a \in \mathcal{A}} \tilde{\pi}_{\mathcal{B}}(a_i|s_i) \cdot \exp(\beta \cdot [\tilde{\mathcal{T}}V^* - V^*](s,a))}{\sum_{a \in \mathcal{A}} \pi_{\mathcal{B}}(a_i|s_i) \cdot \exp(\beta \cdot [\tilde{\mathcal{T}}V^* - V^*](s,a))}. \tag{40}$$

By the definition of corruption levels, we have

$$\frac{\tilde{\pi}_{\mathrm{E}}(a_i|s_i)}{\pi_{\mathrm{E}}(a_i|s_i)} \leq \zeta_i' \cdot \exp(\beta\zeta_i) \cdot \frac{\zeta_i' \cdot \exp(\beta\zeta_i) \sum_{a \in \mathcal{A}} \tilde{\pi}_{\mathcal{B}} \cdot \exp(\beta \cdot [\tilde{\mathcal{T}}V^* - V^*](s,a))}{\tilde{\pi}_{\mathcal{B}} \cdot \exp(\beta \cdot [\tilde{\mathcal{T}}V^* - V^*](s,a))} \tag{41}$$

$$= \zeta_i'^2 \cdot \exp(2\beta\zeta_i). \tag{42}$$

Thus, we can derive

$$W_1(p^{\tilde{\pi}_E}, p^{\pi_E}) \leq \frac{2Mr_{\max}}{(1-\gamma)^2} \sqrt{1 - e^{-\frac{M\zeta}{N}}}. \tag{43}$$

Then, for $W_1(p^{\pi_E}, q^{\pi_{IQL}})$, we have

$$W_1(p^{\pi_E}, q^{\pi_{IQL}}) \leq \frac{Mr_{\max}}{(1-\gamma)^2} \mathbb{E}_{s \sim d^{\pi_E}} ||\pi_E(\cdot|s) - \pi_{IQL}(\cdot|s)||_1 \leq \frac{2Mr_{\max}}{(1-\gamma)^2} \sqrt{1 - e^{-\epsilon_1}}, \tag{44}$$

where $\epsilon_1 = \mathbb{E}_{s \sim b} [\mathcal{D}_{\mathrm{KL}}(\pi_{\mathrm{E}}(\cdot|s) \| \pi_{\mathrm{IQL}}(\cdot|s))]$.

Finally, for $W_1(q^{\tilde{\pi}_{IQL}}, p^{\tilde{\pi}_E})$, we have

$$W_1(q^{\tilde{\pi}_{IQL}}, p^{\tilde{\pi}_E}) \leq \frac{Mr_{\max}}{(1-\gamma)^2} \mathbb{E}_{s \sim d^{\tilde{\pi}_E}} ||\tilde{\pi}_E(\cdot|s) - \tilde{\pi}_{IQL}(\cdot|s)||_1 \leq \frac{2Mr_{\max}}{(1-\gamma)^2} \sqrt{1 - e^{-\epsilon_2}}. \tag{45}$$

Therefore, we have

$$W_1(q^{\tilde{\pi}}, q^{\pi}) \le \frac{2Mr_{\max}}{(1-\gamma)^2}(\sqrt{1 - e^{-\frac{M\zeta}{N}}} + \sqrt{1 - e^{-\epsilon_1}} + \sqrt{1 - e^{-\epsilon_2}}). \tag{46}$$

$\square$

Note that in Theorem A.4, the major difference between TRACER and IQL/RIQL is that TRACER uses the action-value and value distributions rather than the action-value and value functions in IQL/RIQL. Therefore, we provide this theorem to prove an upper bound on the difference in value distributions of TRACER to show its robustness, which also provides a guarantee of how TRACER's performance degrades with increased data corruptions.

## A.2 Derivation of Loss Functions

For Equations (7), (8), and (9), we provide the detailed derivations.

Firstly, for Equation (7), we maximize the posterior and have

$$\begin{aligned}
\max \log p(D) &= \max \log \mathbb{E}_{S_t, R_t}\left[p(D|S_t, R_t)\right] \\
&\ge \max \mathbb{E}_{S_t, R_t}\left[\log p(D|S_t, R_t)\right] \\
&= \max \mathbb{E}_{S_t, R_t}\left[\int_{\mathcal{P}(\mathcal{A})} p_{\psi_a}(A_t|D, S_t, S_{t+1}) \cdot \log p(D|S_t, R_t)dA_t\right].
\end{aligned} \tag{47}$$

$$\begin{aligned}
\log p(D|S_t, R_t) &= \int_{\mathcal{P}(\mathcal{A})} p_{\psi_a}(A_t|D, S_t, S_{t+1}) \cdot \log \frac{p(D, A_t|S_t, R_t, S_{t+1}) \cdot p_{\psi_a}(A_t|D, S_t, S_{t+1})}{p_{\psi_a}(A_t|D, S_t, S_{t+1}) \cdot p(A_t|D, S_t, S_{t+1})}dA_t \\
&= \int_{\mathcal{P}(\mathcal{A})} p_{\psi_a}(A_t|D, S_t, S_{t+1}) \cdot \log \frac{p(D, A_t|S_t, R_t)}{p_{\psi_a}(A_t|D, S_t, S_{t+1})}dA_t \\
&\qquad + \int_{\mathcal{P}(\mathcal{A})} p_{\psi_a}(A_t|D, S_t, S_{t+1}) \cdot \log \frac{p_{\psi_a}(A_t|D, S_t, S_{t+1})}{p(A_t|D, S_t, S_{t+1})}dA_t \\
&= \int_{\mathcal{P}(\mathcal{A})} p_{\psi_a}(A_t|D, S_t, S_{t+1}) \cdot \log \frac{p(D, A_t|S_t, R_t)}{p_{\psi_a}(A_t|D, S_t, S_{t+1})}dA_t \\
&\qquad + \mathcal{D}_{\mathrm{KL}}\left(p_{\psi_a}(A_t|D, S_t, S_{t+1})\|p(A_t|D, S_t, S_{t+1})\right) \\
&= L_b + \mathcal{D}_{\mathrm{KL}}\left(p_{\psi_a}(A_t|D, S_t, S_{t+1})\|p(A_t|D, S_t, S_{t+1})\right) \tag{48} \\
&\ge L_b. \tag{49}
\end{aligned}$$

Note that $L_b$ is

$$\begin{aligned}
L_b &= \int_{\mathcal{P}(\mathcal{A})} p_{\psi_a}(A_t|D, S_t, S_{t+1}) \cdot \log \frac{p(D, A_t|S_t, R_t)}{p_{\psi_a}(A_t|D, S_t, S_{t+1})}dA_t \tag{50} \\
&= \int_{\mathcal{P}(\mathcal{A})} p_{\psi_a}(A_t|D, S_t, S_{t+1}) \log \frac{\pi(A_t|S_t) \cdot p_\theta(D|S_t, A_t, R_t)}{p_{\psi_a}(A_t|D, S_t, S_{t+1})}dA_t \\
&= -\mathcal{D}_{\mathrm{KL}}\left(p_{\psi_a}(A_t|D, S_t, S_{t+1})\|\pi(A_t|S_t)\right) \\
&\qquad + \int_{\mathcal{P}(\mathcal{A})} p_{\psi_a}(A_t|D, S_t, S_{t+1}) \log p_\theta(D|S_t, A_t, R_t)dA_t.
\end{aligned}$$

Secondly, for Equation (8), we have

$$\begin{aligned}
\max \log p(D) &= \max \log \mathbb{E}_{S_t, A_t} \left[ P(D|S_t, A_t) \right] \\
&\geq \max \mathbb{E}_{S_t, A_t} \left[ \log P(D|S_t, A_t) \right] \\
&= \max \mathbb{E}_{S_t, A_t} \left[ \int_{\mathcal{P}(\mathcal{R})} p_{\psi_r}(R_t|D, S_t, A_t) \cdot \log p(D|S_t, A_t) dR_t \right] \\
&\geq \max \mathbb{E}_{S_t, A_t} \left[ - \mathcal{D}_{\text{KL}} \left( p_{\psi_r}(R_t|D, S_t, A_t) \| p(R_t|S_t, A_t) \right) \right. \\
&\qquad\qquad \left. + \int_{\mathcal{P}(\mathcal{R})} p_{\psi_r}(R_t|D, S_t, A_t) \log p(D|S_t, A_t, R_t) dR_t \right]. \quad (51)
\end{aligned}$$

Finally, for Equation (9), we have

$$\begin{aligned}
\max \log p(D) &= \max \log \mathbb{E}_{A_t, R_t} \left[ p(D|A_t, R_t) \right] \\
&\geq \max \mathbb{E}_{A_t, R_t} \left[ \log p(D|A_t, R_t) \right] \\
&= \max \mathbb{E}_{A_t, R_t} \left[ \int_{\mathcal{P}(\mathcal{S})} p_{\psi_s}(S_t|D, A_t, R_t) \cdot \log p(D|A_t, R_t) dS_t \right] \\
&\geq \max \mathbb{E}_{A_t, R_t} \left[ - \mathcal{D}_{\text{KL}} \left( p_{\psi_s}(S_t|D, A_t, R_t) \| p(S_t|A_t, R_t) \right) \right. \\
&\qquad\qquad \left. + \int_{\mathcal{P}(\mathcal{S})} p_{\psi_s}(S_t|D, A_t, R_t) \log p(D|S_t, A_t, R_t) dS_t \right]. \quad (52)
\end{aligned}$$

Therefore, we have Equations (7), (8), and (9).

### A.3 Estimation of Entropy

We estimate differential entropy following [52]. We have differential entropy as follows. Note that we omit the condition in the following equations.

$$\mathcal{H}(p_\theta(D)) = \mathbb{E}[-p_\theta(D)] = - \int_{\mathbb{R}} p_\theta(D) \log p(D) dx. \quad (53)$$

Then, we consider a continuous function $p_\theta$ discretized into bins of size $\Delta$. By the mean-value theorem there exists a action value $D_i$ in each bin such that

$$p_\theta(D_i)\Delta = \int_{i\Delta}^{(i+1)\Delta} p_\theta(D) dD. \quad (54)$$

We can estimate the integral of $p_\theta$ (in the Riemannian sense) by

$$\int_{-\infty}^{\infty} p_\theta(D) dD = \lim_{\Delta \to 0} \sum_{i=-\infty}^{\infty} p_\theta(D_i)\Delta. \quad (55)$$

Thus, we have

$$\begin{aligned}
\mathcal{H}(p_\theta(D)) = \int_{-\infty}^{\infty} p_\theta(D) \log p(D) dx &= \lim_{\Delta \to 0} \left( \mathcal{H}_\Delta(p_\theta(D)) + \log \Delta \right) \\
&= - \lim_{\Delta \to 0} \sum_{i=-\infty}^{\infty} p_\theta(D_i)\Delta \log p_\theta(D_i). \quad (56)
\end{aligned}$$

Based on Equation (56), we can derive Equation (15), where $\hat{\varsigma}_n$ denotes the midpoint of $p_\theta(D_i)$, and $\overline{D}_{\theta_i}^{\varsigma_n}$ denotes $\Delta$.

Table 4: Notations used in our proofs.

| Notations | Descriptions |
| --- | --- |
| $\zeta$ | Cumulative corruption level. |
| $\zeta_i$ | Metric quantifying the corruption level of rewards and next states (transition dynamics). |
| $\zeta_i'$ | Metric quantifying the corruption level of states and actions. |
| $\pi_b(\cdot\|s)$ | The behavior policy that is used to collect clean data. |
| $\pi_{\mathcal{B}}(\cdot\|s)$ | The behavior policy that is used to collect corrupted data. |
| $\pi_E(\cdot\|s)$ | The policy that we want to learn under clean data. |
| $\tilde{\pi}_E(\cdot\|s)$ | The policy that we are learning under corrupted data. |
| $\pi_{IQL}(\cdot\|s)$ | The learned policy using IQL's weighted imitation learning under clean data. |
| $\tilde{\pi}_{IQL}(\cdot\|s)$ | The learned policy using IQL's weighted imitation learning under corrupted data. |
| $d^\pi(s,a)$ | The probability density function associated with policy $\pi$ at state $s$ and action $a$. |
| $W_1(p,q)$ | The Wasserstein-1 distance that measures the difference between distributions $p$ and $q$. |
| $q^\pi$ | The value distribution of the policy $\pi$. |
| $Z^\pi(s)$ | The random variable of the value distribution $q^\pi(\cdot\|s)$. |
| $\epsilon_1$ | KL divergence between $\pi_E$ and $\pi_{\text{IQL}}$, i.e., the standard imitation error under clean data. |
| $\epsilon_2$ | KL divergence between $\tilde{\pi}_E$ and $\tilde{\pi}_{IQL}$, i.e., the standard imitation error under corrupted data. |
| $\hat{\varsigma}_n$ | The midpoint of the probability density $p_\theta$. |

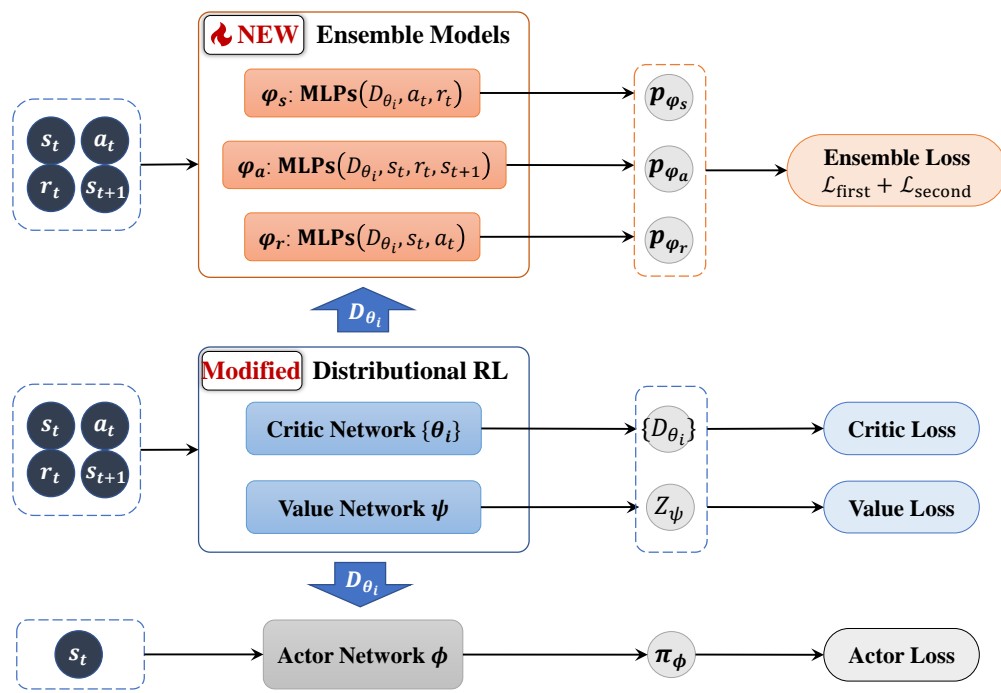

Figure 4: Architecture of TRACER.

# B  TRACER Approach

## B.1  Architecture of TRACER

On the actor-critic architecture based on IQL, our approach TRACER adds just one ensemble model $(p_{\varphi_a}, p_{\varphi_r}, p_{\varphi_s})$ to reconstruct the data distribution and replace the function approximation in the critic with a distribution estimation (see Figure 4). Based on our experiments, this structure significantly improves the ability to handle both simultaneous and individual corruptions.

## B.2 Implementation Details for TRACER

TRACER consists of four components: the actor network, the critic network, the value network, and the observation model. We first implement the actor network for decision-making with a 3-layer MLP, where the dimension of the hidden layers are 256 and the activation functions are ReLUs. Then, we utilize quantile regression [44] to design the critic networks and the value network, approximating the distributions of the action-value function $D$ and value function $Z$, respectively.

Specifically, for the action-value distribution, we use a function $h : [0,1] \rightarrow \mathbb{R}^d$ to compute an embedding for the sampling point $\tau$. Then, we can obtain the corresponding approximation by $D^\tau(s,a) \approx f(g(s) \odot h(\tau))_a$. Here, $g : \mathcal{S} \rightarrow \mathbb{R}^d$ is the function computed by MLPs and $f : \mathbb{R}^d \rightarrow \mathbb{R}^{|\mathcal{A}|}$ is the subsequent fully-connected layers mapping $g(s)$ to the estimated action-values $D(s,a) \approx f(g(s))_a$. For the approximation of the value distribution $Z^\tau(s)$, we leverage the same architecture. Note that the function $h$ is implemented by a fully-connected layer, and the output dimension is set to 256. We then use an ensemble model with $K$ networks. Each network is a 3-layer MLP, consisting of 256 units and ReLU activations. By using the ensemble model, we can construct the critic network, using the quantile regression for optimization. Moreover, the value network is constructed by a 3-layer MLP. The dimension of the hidden layers are also 256 and activation functions are ReLUs.

The observation model uses an ensemble model with 3 neural networks. Each network is used to reconstruct different observations in the offline dataset. It consists of two fully connected layers, and the activation functions are ReLUs. We apply a masked matrix during the learning process for updating the observation model. Thus, each network in the observation model can use different input data to compute Equations (7), (8), and (9), respectively.

Following the setting of [18], we list the hyper-parameters of $N$, $K$, and $\alpha$ for the approxmiation of action-value functions, and $\kappa$ in the Huber loss in TRACER under random and adversarial corruption in Table 5 and Table 6, respectively. Here, $N$ is the number of samples $\tau$, and $K$ is the number of ensemble models.

Moreover, we apply Adam optimizer using a learning rate $1 \times 10^{-3}$, $\gamma = 0.99$, target smoothing coefficient $\tau = 0.05$, batch size 256 and the update frequency for the target network is 2. The corruption rate is $c = 0.3$ and the corruption scale is $\epsilon = 1.0$ for all experiments. For the loss function of observation models, we use a linear decay parameter $\eta$ from 0.0001 to 0.01 to trade off the loss $\mathcal{L}_{\text{first}}$ and $\mathcal{L}_{\text{second}}$. We update each algorithm for 3000 epochs. Each epoch uses 1000 update times following [56]. Then, we evaluate each algorithm in clean environments for 100 epochs and report the average normalized return (calculated by $100 \times \frac{\text{score} - \text{random score}}{\text{expert score} - \text{random score}}$ ) over four random seeds.

Table 5: Hyper-parameters used for TRACER under the random corruption benchmark.

| Environments | Corruption Types | $N$ | $K$ | $\alpha$ | $\kappa$ |
|---|---|---|---|---|---|
| Halfcheetah | observation | 32 | 5 | 0.25 | 0.1 |
| | action | 32 | 5 | 0.25 | 0.1 |
| | reward | 32 | 5 | 0.25 | 0.1 |
| | dynamics | 32 | 5 | 0.25 | 0.1 |
| | simultaneous | 32 | 5 | 0.25 | 0.1 |
| Walker2d | observation | 32 | 5 | 0.25 | 0.1 |
| | action | 32 | 5 | 0.25 | 0.1 |
| | reward | 32 | 5 | 0.5 | 0.1 |
| | dynamics | 32 | 5 | 0.25 | 1.0 |
| | simultaneous | 32 | 5 | 0.25 | 0.1 |
| Hopper | observation | 32 | 5 | 0.25 | 0.1 |
| | action | 32 | 5 | 0.25 | 0.1 |
| | reward | 32 | 5 | 0.5 | 0.1 |
| | dynamics | 32 | 5 | 0.25 | 0.1 |
| | simultaneous | 32 | 5 | 0.25 | 0.1 |
| CARLA | simultaneous | 32 | 5 | 0.25 | 0.1 |

Table 6: Hyper-parameters used for TRACER under the adversarial corruption benchmark.

| Environments | Corruption Types | $N$ | $K$ | $\alpha$ | $\kappa$ |
|---|---|---|---|---|---|
| Halfcheetah | observation | 32 | 5 | 0.1 | 0.1 |
| | action | 32 | 5 | 0.1 | 0.1 |
| | reward | 32 | 5 | 0.1 | 0.1 |
| | dynamics | 32 | 5 | 0.1 | 0.1 |
| | simultaneous | 32 | 5 | 0.25 | 0.1 |
| Walker2d | observation | 32 | 5 | 0.25 | 1.0 |
| | action | 32 | 5 | 0.1 | 1.0 |
| | reward | 32 | 5 | 0.1 | 0.1 |
| | dynamics | 32 | 5 | 0.25 | 0.1 |
| | simultaneous | 32 | 5 | 0.25 | 0.1 |
| Hopper | observation | 32 | 5 | 0.25 | 0.5 |
| | action | 32 | 5 | 0.25 | 0.1 |
| | reward | 32 | 5 | 0.5 | 0.1 |
| | dynamics | 32 | 5 | 0.5 | 0.1 |
| | simultaneous | 32 | 5 | 0.5 | 0.001 |

## C  Detailed Experiments

### C.1  Details of Data Corruptions

We follow the corruption settings proposed by [18], applying either random or adversarial corruptions to each element of the offline data, namely state, action, reward, and dynamics (or "next-state"), to simulate potential attacks that may occur during the offline data collection and usage process in real-world scenarios. We begin with the details of random corruption below.

- **Random observation corruption.** We randomly sample $c\%$ transitions $(s, a, r, s')$ from the offline dataset, and for each of these selected states $s$, we add noise to form $\hat{s} = s + \lambda \cdot std(s)$, where $\lambda \sim Uniform[-\epsilon, \epsilon]^{d_s}$, $c$ is the corruption rate, $\epsilon$ is the corruption scale, $d_s$ refers to the dimension of states and $std(s)$ represents the $d_s$-dimensional standard deviation of all states in the offline dataset.

- **Random action corruption.** We randomly sample $c\%$ transitions $(s, a, r, s')$ from the offline dataset, and for each of these selected actions $a$, we add noise to form $\hat{a} = a + \lambda \cdot std(a)$, where $\lambda \sim Uniform[-\epsilon, \epsilon]^{d_a}$, $d_a$ refers to the dimension of actions and $std(a)$ represents the $d_a$-dimensional standard deviation of all actions in the offline dataset.

- **Random reward corruption.** We randomly sample $c\%$ transitions $(s, a, r, s')$ from the offline dataset, and for each of these selected rewards $r$, we modify it to $\hat{r} = Uniform[-30 \cdot \epsilon, 30 \cdot \epsilon]$. We adopt this harder reward corruption setting since [81] has found that offline RL algorithms are often insensitive to small perturbations of rewards.

- **Random dynamics corruption.** We randomly sample $c\%$ transitions $(s, a, r, s')$ from the offline dataset, and for each of these selected next-step states $s'$, we add noise to form $\hat{s'} = s' + \lambda \cdot std(s')$, where $\lambda \sim Uniform[-\epsilon, \epsilon]^{d_{s'}}$, $d_{s'}$ refers to the dimension of next-step states and $std(s')$ represents the $d_{s'}$-dimensional standard deviation of all next states in the offline dataset.

The harder adversarial corruption settings are detailed as follows:

- **Adversarial observation corruption.** To impose adversarial attack on the offline dataset, we first need to pretrain an EDAC agent with a set of $Q_p$ functions and a policy function $\pi_p$ using clean dataset. Then, we randomly sample $c\%$ transitions $(s, a, r, s')$ from the offline dataset, and for each of these selected states $s$, we attack it to form $\hat{s} = min_{\hat{s} \in \mathbb{B}_d(s, \epsilon)} Q_p(\hat{s}, a)$. Here, $\mathbb{B}_d(s, \epsilon) = \{\hat{s} | |\hat{s} - s| \leq \epsilon \cdot std(s)\}$ regularizes the maximum difference for each state dimension.

- **Adversarial action corruption.** We randomly sample $c\%$ transitions $(s, a, r, s')$ from the offline dataset, and for each of these selected actions $a$, we use the pretrained EDAC agent to attack it to form $\hat{a} = min_{\hat{a} \in \mathbb{B}_d(a,\epsilon)} Q_p(s, \hat{a})$. Here, $\mathbb{B}_d(a, \epsilon) = \{\hat{a} || \hat{a} - a| \leq \epsilon \cdot std(a)\}$ regularizes the maximum difference for each action dimension.
- **Adversarial reward corruption.** We randomly sample $c\%$ transitions $(s, a, r, s')$ from the offline dataset, and for each of these selected rewards $r$, we directly attack it to form $\hat{r} = -\epsilon \times r$ without any adversarial model. This is because that the objective of adversarial reward corruption is $\hat{r} = \min_{\hat{r} \in \mathbb{B}(r,\epsilon)} \hat{r} + \gamma \mathbb{E}[Q(s', a')]$. Here $\mathbb{B}(r, \epsilon) = \{\hat{r} \mid |\hat{r} - r| \leq (1+\epsilon) \cdot r_{\max}\}$ regularizes the maximum distance for rewards. Therefore, we have $\hat{r} = -\epsilon \times r$.
- **Adversarial dynamics corruption.** We randomly sample $c\%$ transitions $(s, a, r, s')$ from the offline dataset, and for each of these selected next-step states $s'$, we use the pretrained EDAC agent to attack it to form $\hat{s}' = min_{\hat{s}' \in \mathbb{B}_d(s',\epsilon)} Q_p(\hat{s}', \pi_p(s'))$. Here, $\mathbb{B}_d(s', \epsilon) = \{\hat{s}' || \hat{s}' - s'| \leq \epsilon \cdot std(s')\}$ regularizes the maximum difference for each dimension of the dynamics.

The optimization of the above adversarial corruptions are implemented through Projected Gradient Descent [53, 54]. Taking adversarial observation corruption for example, we first initialize a learnable vector $z \in [-\epsilon, \epsilon]^{d_s}$, and then conduct a 100-step gradient descent with a step size of 0.01 for $\hat{s} = s + z \cdot std(s)$, and clip each dimension of $z$ within the range $[-\epsilon, \epsilon]$ after each update. For action and dynamics corruption, we conduct similar operation.

Building upon the corruption settings for individual elements as previously discussed, we further intensify the corruption to simulate the challenging conditions that might be encountered in real-world scenarios. We present the details of the simultaneous corruption below:

- **Random simultaneous corruptions.** We sequentially conduct random observation corruption, random action corruption, random reward corruption, and random dynamics corruption to the offline dataset in order. That is to say, we randomly select $c\%$ of the transitions each time and corrupt one element among them, repeating four times until states, actions, rewards, and dynamics of the offline dataset are all corrupted.
- **Adversarial simultaneous corruptions.** We sequentially conduct adversarial observation corruption, adversarial action corruption, adversarial reward corruption, and adversarial dynamics corruption to the offline dataset in order. That is to say, we randomly select $c\%$ of the transitions each time and attack one element among them, repeating four times until states, actions, rewards, and dynamics of the offline dataset are all corrupted.

## C.2 Details for CARLA

We conduct experiments in CARLA from D4RL benchmark. We use the clean environment 'CARLA-Lane-v0' to evaluate IQL, RIQL, and TRACER (Ours). We report each mean result with the standard deviation in the left of Figure 2 over four random seeds for 3000 epochs. We apply the random simultaneous data corruptions, where each element in the offline dataset (including states, actions, rewards, and next states) may involve random noise. We follow the setting in Section 4.2, using the corruption rate $c = 0.3$ and scale $\epsilon = 1.0$ in the CARLA task. The results in the left of Figure 2 show the superiority of TRACER in the random simultaneous corruptions. We provide the hyperparameters of TRACER in Table 5.

## C.3 Additional Experiments and Analysis

### C.3.1 Results Comparison between TRACER and RIQL under Individual Corruptions

In Tables 2 and 3 of the main text, we adhered to commonly used settings for individual corruptions in corruption-robust offline RL. We directly followed hyperparameters from RIQL (i.e., $\kappa$ for huber loss, the ensemble number $K$, and $\alpha$ in action-value functions, see Tables 5 and 6). Results show that TRACER outperforms RIQL in **18 out of 24** settings, demonstrating its robustness even when aligned with RIQL's hyperparameters.

Further, we explore hyperparameter tuning, specifically of $\kappa$, on Hopper task to improve TRACER's performance. This results in TRACER outperforming RIQL in **7 out of 8** settings on Hopper, up

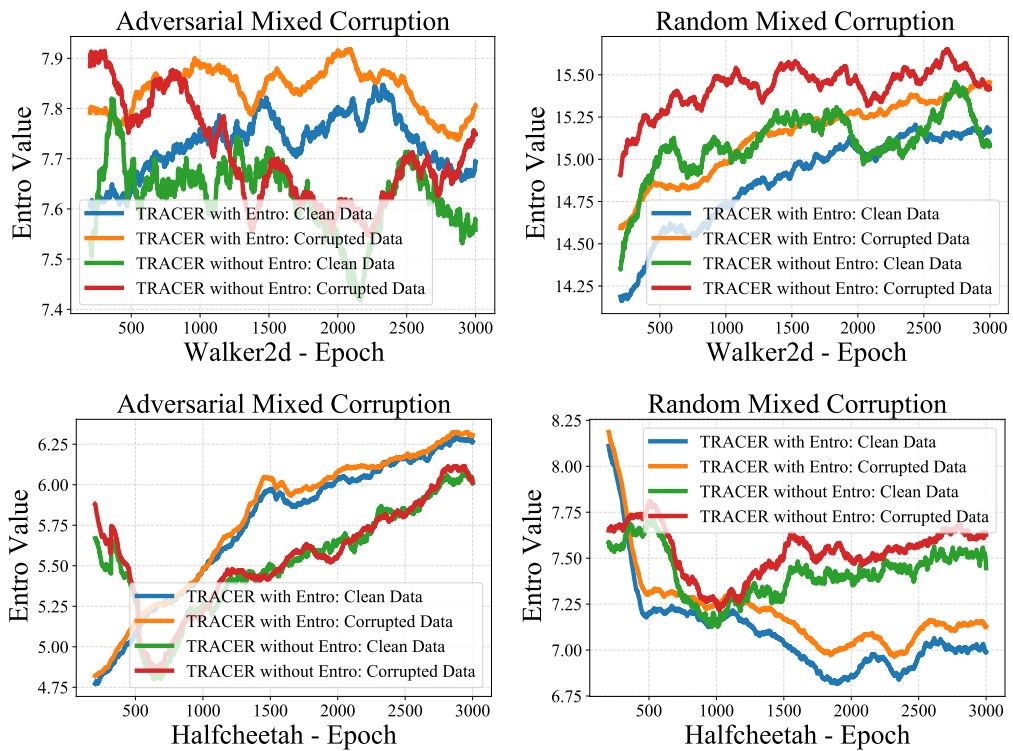

Figure 5: We report the smoothed curves of mean of entropy values for each batch in 'Walker2d-medium-replay-v2' and 'Halfcheetah-medium-replay-v2' under adversarial and random simultaneous data corruptions.

Table 7: Average results and standard errors with 2 seeds and 64 batch sizes in Hopper-medium-replay-v2 task for hyperparameter tuning.

|  | Random Dynamics $\kappa = 0.01$ | Random Dynamics $\kappa = 0.1$ | Random Dynamics $\kappa = 0.5$ | Random Dynamics $\kappa = 1.0$ |
|---|---|---|---|---|
| TRACER (Ours) | $52.6 \pm 0.9$ | $\mathbf{57.5 \pm 13.0}$ | $45.1 \pm 10.5$ | $44.0 \pm 6.5$ |
|  | Adversarial Reward $\kappa = 0.01$ | Adversarial Reward $\kappa = 0.1$ | Adversarial Reward $\kappa = 0.5$ | Adversarial Reward $\kappa = 1.0$ |
| TRACER (Ours) | $\mathbf{82.2 \pm 0.1}$ | $61.3 \pm 0.8$ | $56.7 \pm 1.6$ | $51.3 \pm 2.1$ |

from 5 settings (see Tables 7 and 8). The further improvement highlights TRACER's potential to achieve greater performance gains.

Based on Table 7, we find that TRACER requires low $\kappa$ in Huber loss, using L1 loss for large errors. Thus, TRACER can linearly penalize corrupted data and reduce its influence on the overall model fit.

### C.3.2 Additional Experiments

We further conduct experiments on two AntMaze datasets and two additional Mujoco datasets, presenting results under random simultaneous corruptions in Table 9.

Each result represents the mean and standard error over four random seeds and 100 episodes in clean environments. For each experiment, the methods train agents using batch sizes of 64 for 3000 epochs. Building upon RIQL, we apply the experiment settings as follows. (1) For the two Mujoco datasets, we use a corruption rate of $c = 0.3$ and scale of $\epsilon = 1.0$. Note that simultaneous corruptions with $c = 0.3$ implies that approximately $76.0\%$ of the data is corrupted. (2) For the two AntMaze datasets, we use the corruption rate of 0.2, corruption scales for observation (0.3), action (1.0), reward (30.0), and dynamics (0.3).

Table 8: Average results and standard errors with 2 seeds and 64 batch sizes in Hopper-medium-replay-v2 task under individual corruptions.

| | Random Observation | Random Action | Random Reward | Random Dynamics |
|---|---|---|---|---|
| RIQL | $62.4 \pm 1.8$ | $90.6 \pm 5.6$ | $84.8 \pm 13.1$ | $51.5 \pm 8.1$ |
| TRACER (Raw) | $\mathbf{62.7 \pm 8.2}$ | $\mathbf{92.8 \pm 2.5}$ | $\mathbf{85.7 \pm 1.4}$ | $49.8 \pm 5.3$ |
| TRACER (New) | - | - | - | $\mathbf{53.8 \pm 13.5}$ |
| | Adversarial Observation | Adversarial Action | Adversarial Reward | Adversarial Dynamics |
| RIQL | $50.8 \pm 7.6$ | $63.6 \pm 7.3$ | $65.8 \pm 9.8$ | $\mathbf{65.7 \pm 21.1}$ |
| TRACER (Raw) | $\mathbf{64.5 \pm 3.7}$ | $\mathbf{67.2 \pm 3.8}$ | $64.3 \pm 1.5$ | $61.1 \pm 6.2$ |
| TRACER (New) | - | - | $\mathbf{71.7 \pm 5.3}$ | - |

Table 9: The average scores and standard errors under random simultaneous corruptions.

| | AntMaze-Medium-Play-v2 | AntMaze-Medium-Diverse-v2 | Walker2d-Medium-Expert-v2 | Hopper-Medium-Expert-v2 |
|---|---|---|---|---|
| IQL | $0.0 \pm 0.0$ | $0.0 \pm 0.0$ | $20.6 \pm 3.4$ | $1.4 \pm 0.3$ |
| RIQL | $0.0 \pm 0.0$ | $0.0 \pm 0.0$ | $23.6 \pm 4.3$ | $3.9 \pm 1.8$ |
| TRACER | $\mathbf{7.5 \pm 3.7}$ | $\mathbf{6.6 \pm 1.4}$ | $\mathbf{47.0 \pm 6.6}$ | $\mathbf{55.5 \pm 2.9}$ |

The results in Table 9 show that TRACER significantly outperforms other methods in **all these tasks** with the aforementioned AntMaze and Mujoco datasets.

## C.4 Details for Evaluation and Ablation Studies

### C.4.1 Evaluation for Robustness of TRACER across Different Scales of Corrupted Data

Theorem A.4 shows that the higher the scale of corrupted data, the greater the difference in action-value distributions and the lower the TRACER's performance. Thus, we further conduct experiments to evaluate TRACER across various corruption levels. Specifically, we apply different corruption rate $c\%$ in all four elements of the offline dataset. We randomly select $c\%$ of transitions from the dataset and corrupt one element in each selected transition. Then, we repeat this step four times until all elements are corrupted. Therefore, approximately $100 \cdot (1 - (1 - c)^4)\%$ of data in the offline dataset is corrupted.

In Table 10, we evaluate TRACER using different $c\%$, including $10\%, 20\%, 30\%, 40\%,$ and $50\%$. These rates correspond to approximately $34.4\%, 59.0\%, 76.0\%, 87.0\%,$ and $93.8\%$ of the data being corrupted. The results in Table 10 demonstrate that while TRACER is robust to simultaneous corruptions, its scores depend on the extent of corrupted data it encounters.

### C.4.2 Details for Evaluation of the Entropy-based Uncertainty Measure

In Figure 5, we additionally report the entropy values of TRACER with and without using entropy-based uncertainty measure, corresponding to Figure 3. The curves illustrate that TRACER using entropy-based uncertainty measure can effectively regulate the loss associated with corrupted data, reducing the influence of corrupted samples. Therefore, the estimated entropy values of corrupted data can be higher than those of clean data.

### C.4.3 Ablation Study for Bayesian Inference

We conduct experiments of TRACER and TRACER without Bayesian inference, i.e., RIQL combined with distributional RL methods, namely DRIQL. The experiments are on 'Walker2d-medium-replay-v2' under random simultaneous data corruptions. We employ the same hyperparameters of Huber loss for these methods, using $\alpha = 0.25$ and $\kappa = 1.0$.

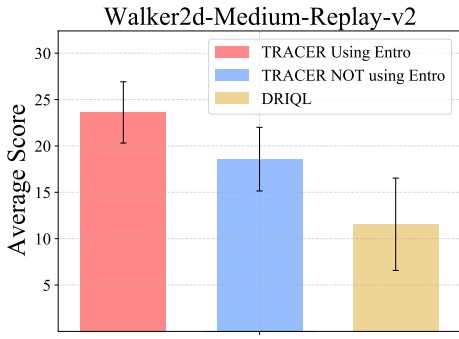

Figure 6: Results and standard errors under random simultaneous corruptions.

We report each mean result with the standard deviation in Figure 6. Each result is averaged over four seeds for 3000 epochs. These results show the effectiveness of our proposed Bayesian inference.

Table 10: Results in Hopper-medium-replay-v2 under various random simultaneous corruption levels.

| Corrupt rate $c$ corrupted data/all data | 0.1 $\approx 34.4\%$ | 0.2 $\approx 59.0\%$ | 0.3 $\approx 76.0\%$ | 0.4 $\approx 87.0\%$ | 0.5 $\approx 93.8\%$ |
|---|---|---|---|---|---|
| RIQL | $43.9 \pm 10.3$ | $55.9 \pm 5.7$ | $22.5 \pm 10.0$ | $21.6 \pm 6.2$ | $15.6 \pm 1.8$ |
| TRACER (Ours) | $\mathbf{79.6 \pm 3.5}$ | $\mathbf{64.0 \pm 3.0}$ | $\mathbf{28.8 \pm 7.1}$ | $\mathbf{25.9 \pm 2.2}$ | $\mathbf{19.7 \pm 1.2}$ |

## D    Compute Resource

In this subsection, we provide the computational cost of our approach TRACER.

For the MuJoCo tasks, including Halfcheetah, Walker2d, and Hopper, the average training duration is 40.6 hours. For the CARLA task, training extends to 51 hours. We conduct all experiments on NVIDIA GeForce RTX 3090 GPUs.

To compare the computational cost, we report the average epoch time on Hopper in Table 11, where results of baselines (including DT [82]) are from [18]. The formula for computational cost is:

$$\frac{\text{avg\_epoch\_time\_of\_RIQL\_in\_[18]}}{\text{avg\_epoch\_time\_we\_run\_RIQL}} \times \text{avg\_epoch\_time\_we\_run\_TRACER}.$$

Note that TRACER requires a long epoch time due to two main reasons:

1. Unlike RIQL and IQL, which learn one-dimensional action-value functions, TRACER generates multiple samples for the estimation of action-value distributions. Following [43], we generate 32 samples of action values for each state-action pair.

2. TRACER uses states, actions, and rewards as observations to update models, estimating the posterior of action-value distributions.

Table 11: Average epoch time.

| Algorithm | BC | DT | EDAC | MSG | CQL | IQL | RIQL | TRACER (Ours) |
|---|---|---|---|---|---|---|---|---|
| Times (s) | 3.8 | 28.9 | 14.1 | 12.0 | 22.8 | 8.7 | 9.2 | 19.4 |

## E    More Related Work

**Online RL.** In general, standard online RL fall into two categories: model-free RL [83–85] and model-based RL [86, 71]. In recent years, RL has achieved great success in many important real-world decision-making tasks [87–93]. However, the online RL methods still typically rely on active data collection to succeed, hindering their application in scenarios where real-time data collection is expensive, risky, and/or impractical. Thus, we focus on the offline RL paradigm in this paper.

## F    Code

We implement our codes in Python version 3.8 and make the code available online [2].

## G    Limitations and Negative Societal Impacts

TRACER's performance is related to the extent of corrupted data within the offline dataset. Although TRACER consistently outperforms RIQL even under conditions of extensive corrupted data (see Table 10), its performance does degrade as the corruption rate (i.e., the extent/scale of corrupted data) increases. To tackle this problem, we look forward to the continued development and optimization of corruption-robust offline RL with large language models, introducing the prior knowledge of clean data against large-scale or near-total corrupted data. Thus, this corruption-robust offline RL can perform well even when faced with large-scale or near-total corrupted data in the offline dataset.

---

[2]https://github.com/MIRALab-USTC/RL-TRACER

This paper proposes a novel approach called TRACER to advance the field of robustness in offline RL for diverse data corruptions, enhancing the potential of agents in real-world applications. Although our primary focus is on technical innovation, we recognize the potential societal consequences of our work, as robustness in offline RL for data corruptions has the potential to influence various domains such as autonomous driving and the large language model field. We are committed to ethical research practices and attach great importance to the social implications and ethical considerations in the development of robustness research in offline RL.

