# OpenReview forum: "Uncertainty-based Offline Variational Bayesian Reinforcement Learning for Robustness under Diverse Data Corruptions"
_NeurIPS.cc/2024/Conference — NeurIPS 2024 poster_

### Official Review · Reviewer_vWB5 · 2024-06-12

**Soundness:** 3
**Presentation:** 3
**Contribution:** 3
**Rating:** 7
**Confidence:** 3

**Summary:**

The authors introduce TRACER, a new Bayesian methodology to capture the uncertainty via offline data for robustness against all types of data corruptions. An appealing feature of TRACER is that it can distinguish corrupted data from clean data using an entropy-based uncertainty measure. Experiments are provided to prove the effectiveness of such a methodology.

**Strengths:**

Very clearly stating the problem, and convincing the reader that's a relevant one. Compelling method to address such a problem.

**Weaknesses:**

The authors missed some important references: [1,2] introduce credal Bayesian deep learning, which is a robust way of performing Bayesian inference via Bayesian Neural Networks (where posterior and predictive distributions are approximated via variational inference) taking into account different types of uncertainty (using so-called credal sets, i.e. closed and convex sets of prior and likelihood probabilities), and quantifying them using entropy-based uncertainty measures. In the future, the authors may also look into [3], which seems like a compelling work for the research venue explored in the paper.

[1] https://arxiv.org/abs/2302.09656

[2] https://link.springer.com/chapter/10.1007/978-3-031-57963-9_1#:~:text=In%20their%20seminal%201990%20paper,bound%20to%20hold%20with%20equality

[3] https://arxiv.org/abs/2308.14815

**Questions:**

I'd like to politely ask the authors to expand the related work with the above references [1,2]. In addition, would it be fair to say that, on page 3, the expectation that finds $\pi^\star$ is an integral taken with respect to the joint probability given by all the sources of randomness $P_0, \pi(\cdot \mid s_t), \rho(\cdot \mid s_t,a_t), P(\cdot \mid s_t,a_t)$?

In equation (1), the closed parenthesis ) should go after $a_0=a$.

**Limitations:**

The limitations are explicitly acknowledged by the authors in the conclusion.

---

> ### Author Rebuttal · Authors · 2024-08-07
>
> We appreciate the reviewer’s insightful and constructive feedback. We have carefully addressed these concerns and accordingly revised the manuscript.
> These comments have not only facilitated significant improvements in our manuscript but have also inspired us for further in-depth studies in our future research.
>
> - **Q1**. The authors missed some important references [1,2]. In the future, the authors may also look into [3], which seems like a compelling work for the research venue explored in the paper.
>   - **A1**. Thank you for pointing out these crucial references. The Shannon entropy for the measures of aleatoric and epistemic uncertainties discussed in these works [1,2] provides important insight and support for our method. We will study these references thoroughly and consider incorporating their findings into our future research. We also appreciate the suggestion to explore the recent work in [3], which could further enhance our understanding and methodology. We will include these additions in our revised manuscript.
>
>   [1] Credal Bayesian Deep Learning. 2023.
>
>   [2] A Novel Bayes' Theorem for Upper Probabilities. Epi UAI 2024.
>
>   [3] Distributionally Robust Statistical Verification with Imprecise Neural Networks. 2023.
>
> - **Q2**. In addition, would it be fair to say that, on page 3, the expectation that finds $\pi^*$ is an integral taken with respect to the joint probability given by all the sources of randomness $P_0$, $\pi (\cdot | s_t)$, $\rho (\cdot | s_t, a_t)$, $P(\cdot |s_t, a_t)$?
>   - **A2**. Yes. On page 3, the expectation that determines $\pi^*$ involves an integral taken with respect to the joint probability distribution influenced by all sources of randomness, including $P_0$, $\pi(\cdot|s_t)$, $\rho(\cdot|s_t,a_t)$, and $P(\cdot|s_t,a_t)$ [1,2].
>
>   [1] Trust Region Policy Optimization. ICML 2015.
>
>   [2] Distributional reinforcement learning with quantile regression. AAAI 2018.
>
> - **Q3**. In equation (1), the closed parenthesis ) should go after $a_0=a$.
>   - **A3**. Thank you for catching the typo in Eq. (1). We will carefully correct this in our revision.

---

> > ### Comment · Reviewer_vWB5 · 2024-08-09
> > **Thank you!**
> >
> > Thanks for your answers; I'm happy to keep my score.

---

> > > ### Author Response · Authors · 2024-08-11
> > > **Thanks for your kind support!**
> > >
> > > Dear Reviewer vWB5,
> > >
> > > Thanks for your kind support and further improvements you have suggested for our manuscript! We are committed to incorporating all your valuable suggestions in the final version, if accepted.
> > >
> > > Thank you again for your valuable comments and guidance.
> > >
> > > Best,
> > >
> > > Authors

---

### Official Review · Reviewer_wmJN · 2024-06-27

**Soundness:** 3
**Presentation:** 2
**Contribution:** 2
**Rating:** 6
**Confidence:** 3

**Summary:**

This paper presents a novel approach called TRACER, aimed at addressing the challenge of learning robust policies from offline datasets that are subject to various data corruptions. The key contribution of this work lies in the integration of Bayesian inference to capture uncertainty within the offline data. TRACER models data corruptions as uncertainty in the action-value function and approximates the posterior distribution of this function. Furthermore, it employs an entropy-based uncertainty measure to distinguish between corrupted and clean data, thereby regulating the loss associated with corrupted data to enhance robustness in clean environments. Experimental results indicate that TRACER outperforms SOTA methods in handling both individual and simultaneous data corruptions.

**Strengths:**

- The studied problem is important, and the paper offers a reasonable solution.
- Theoretical guarantees are provided.
- The empirical results show considerable improvement over prior work.

**Weaknesses:**

- Though I like the insight from Variational Bayesian inference, my major concern is that the method is quite complicated, incorporating numerous components and designs such as the distributional value function, variational inference, and entropy-weighted loss function. This complexity makes this method less preferable in practice.

- Moreover, the results in the main tables (Table 2 and Table 3) appear similar to those of RIQL, raising concerns about the necessity of the included components.

- Another issue is the readability and clarity of the writing. For example, Equations 10 and 11 lack necessary explanation and insight. Additionally, the implementation details for $\phi_a, \phi_b, \phi_c$ are missing, leaving readers unsure of how the method works in practice. Furthermore, Equation 10 seems to include $\theta$ as an input unnecessarily, as the right side does not depend on $\theta$.

- The analysis of the experiments is not thorough. For instance, the accuracy of the measurement for corrupted data decreases during training in Figure 3, which is not discussed and seems unreasonable.

**Questions:**

- The proposed method is complicated. Can this method be simplified?

- The results in the main tables (Table 2 and Table 3) appear similar to those of RIQL, raising concerns about the necessity of the included components.

- Another issue is the readability and clarity of the writing discussed above.

- Why does the accuracy of the measurement for corrupted data decreases during training in Figure 3?

- Moreover, I suggest the authors provide the computational cost comparison with prior work.

**Limitations:**

This paper acknowledges its limitations; however, I identified an additional concern regarding the complexity of the proposed method.

---

> ### Author Rebuttal · Authors · 2024-08-07
>
> We appreciate the reviewer's insightful and constructive comments and suggestions. We respond to each comment as follows and sincerely hope that our responses could properly address your concerns. If so, we would deeply appreciate it if you could raise your score. If not, please let us know your further concerns, and we will continue actively responding to the comments and enhancing our submission.
> - **Q1**. The proposed method is complicated. Can this method be simplified?
>   - **A1**. Thanks for your meaningful comment.
>     - On the actor-critic architecture based on IQL, our approach TRACER adds just one ensemble model $(p_{\varphi_a},p_{\varphi_r},p_{\varphi_s})$ to reconstruct the data distribution and repace the function approximation in the critic with a distribution estimation (see Figure G1 in the PDF of our global response). Therefore, the network structure is not complicated, while significantly improving the ability to handle both simultaneous and individual corruptions.
>     - It is worth noting that our primary aim is the development of corruption-robust offline RL techniques for diverse corruptions. To the best of our knowledge, this study introduces Bayesian inference into corruption-robust offline RL for the first time, prioritizing novelty and robust performance in challenging scenarios with simultaneous corruptions.
>     - We thank you for reminding us of the importance of the balance between complexity and usability. In future work, we plan to refine TRACER by estimating uncertainty directly within the representation space. Thus, we can simply learn action-value functions without using distributional RL. This advancement will potentially broaden the applications of TRACER in real-world scenarios.
> - **Q2**. The results in the main tables (Table 2 and Table 3) appear similar to those of RIQL, raising concerns about the necessity of the included components.
>   - **A2**. Thanks for the insightful comment. Please refer to A3 in our global response.
> - **Q3**. Another issue is the readability and clarity of the writing. For example, Equations 10 and 11 lack necessary explanation and insight. Additionally, the implementation details for $\varphi_a$, $\varphi_r$, $\varphi_s$ are missing, leaving readers unsure of how the method works in practice.
>   - **A3**. Please refer to A2 in our global response.
> - **Q4**. Furthermore, Equation 10 seems to include $\theta$ as an input unnecessarily, as the right side does not depend on $\theta$.
>   - **A4**. Here is our explanation.
>     - The goal of Eq. (10) of our main text is to optimize $(\theta, \varphi_a,\varphi_r,\varphi_s)$, thus minimizing the difference between $p_{\varphi_a}(A|D_\theta,S,R,S')$ and $\pi_{\mathcal{B}}(A|S)$, $p_{\varphi_r}(R|D_\theta,S,A)$ and $\rho_{\mathcal{B}}(R|S,A)$, and $p_{\varphi_s}(S|D_\theta,A,R)$ and $p_{\mathcal{B}}(S)$. More details are in A2 of the global response.
>     - In Eq. (10), the input of each $(\mu_{\varphi}, \Sigma_{\varphi})$ includes $D_\theta$ with the parameter $\theta$.
> - **Q5**. The analysis of experiments is not thorough. Why does the accuracy of the measurement for corrupted data decrease during training in Figure 3?
>   - **A5**. Thanks for your meaningful comment.
>     - We apologize for any confusion caused by the unclear description and presentation of the validation experiments in Figure 3 of our main text. To clearly illustrate the measurement accuracy of agents learned by TRACER with respect to uncertainty during training, we further conduct validation experiments on Walker2d-Medium-Replay-v2.
>
>       Specifically, we evaluate the accuracy every 50 epochs over 3000 epochs. For each evaluation, we sample 500 batches to compute the average entropy of corrupted and clean data. Each batch consists of 32 clean and 32 corrupted data. We illustrate the curves over three seeds in Figure G3 of the PDF of our global response, where each point shows how many of the 500 batches have higher entropy for corrupted data than that of clean data.
>     - Figure G3 illustrates an oscillating upward trend of TRACER's measurement accuracy using entropy under simultaneous corruptions, demonstrating that using the entropy-based uncertainty measure can effectively distinguish corrupted data from clean data.
> - **Q6**. Moreover, I suggest the authors provide the computational cost comparison with prior work.
>   - **A6**. Please refer to A1 in the global response.

---

> > ### Comment · Reviewer_wmJN · 2024-08-09
> > **Thanks for the response**
> >
> > My concerns have mostly been addressed, so I have decided to raise my score. The added diagram and clarification on the writing greatly improve understanding. The authors are expected to include these in the revised paper.

---

> > > ### Author Response · Authors · 2024-08-09
> > > **Thanks for your kind support and for helping us improve the paper!**
> > >
> > > Dear Reviewer wmJN,
> > >
> > > Thanks for your kind support and for helping us improve the paper! We are committed to including all additional diagrams and clarifications and will incorporate your suggestions in the final version, if accepted.
> > >
> > > Thank you again for your valuable comments and guidance.
> > >
> > > Best,
> > >
> > > Authors

---

### Official Review · Reviewer_QZTE · 2024-07-07

**Soundness:** 3
**Presentation:** 2
**Contribution:** 2
**Rating:** 6
**Confidence:** 4

**Summary:**

The paper introduces TRACER, a robust offline reinforcement learning (RL) algorithm designed to address the challenges posed by data corruptions in offline datasets. The corruptions can be
realized in form of states, actions, rewards, and dynamics corruption. The proposed methodology uses Bayesian inference to model and mitigate the impact of corrupted data on the action-value function.
By leveraging an entropy-based uncertainty measure, TRACER can distinguish between corrupted and clean data, thereby reducing the influence of the former and enhancing the robustness of the learned policies.
The paper provides thorough evaluation on MujoCo and Carla offline RL datasets.

**Strengths:**

1. The paper proposes an interesting method of down weighting the contribution corrupted data while learning action value function from the offline dataset using entropy as an uncertainty measure.
2. The paper uses a Bayesian approach to learn the action value function from the data which aids learning from uncorrupted elements in the dataset.
3. The authors provide extensive experimental validation across multiple tasks and corruption types, showing mostly consistent performance improvements over other methods. The paper also
reports ablation studies of using entropy showing potential improvement from the proposed methodology.

**Weaknesses:**

1. It is not clear how the final policy is extracted after learning $D_{\theta}$.
2. The paper relies on many hyperparameters. What kind of kernel is used in the gaussian distribution learning in Eq 10? Is there a upper bound on what amount of corruption the method will be able to handle?
3. While the authors consider individual corruption of one element at a time data collected from in a real world system can have simultaneous corruptions. For
example corrupted action due to adversarial noise will lead the system to different state causing simultaneous corruptions.
4. Notational clarity: The paper introduces a lot of notations which is difficult to keep track of. Specially in the following contexts :
a :  The paper introduces LQ (θi), LV(ψ), Lπ(ϕ) in Eq 4, 5 and 6 but does not discuss how they relate to Dθ. Is the θ used for learning Q function and D same?
b : The paper introduces a new notation qπ in Eq 18 called value distribution. Is this the same as action value distribution? RIQL already proves the robustness of IQL policy with respect of data corruption. What is
the objective of Theorem A.3? It will be helpful to add a discussion on the theoretical analysis. Also Providing a table summarizing the notations in appendix will enhance readability of the text.

**Questions:**

1) How relevant is the reward corruption given offline RL already is robust to noise in the reward function as it mimics the actions for datasets with length bias? Please refer to ref 1
2) The authors aim to leverage all elements in the dataset as observations. Do the authors assume that complete traces of the system are available for which the offline policy is being learned?
In situations of dynamic corruption these traces might be different than the system. For example s_t in the corrupted dataset may not be observed in the real system. How are such cases
handled?
3) As the method learns action-value function D from the data while reducing the influence of corrupted data. What amount of clean data needs to be present in the dataset for this method
to work?
4) What is the reason for not using projected gradient descent for the reward corruption? Also could the author provide some understanding of why the method outperforms RIQL under certain settings and why it does not in some cases?
5) Please also respond to the weaknesses.

1. Li, Anqi, et al. "Survival instinct in offline reinforcement learning." Advances in neural information processing systems 36 (2024).

**Limitations:**

Limitations: The authors discuss the limitation in terms of handling realistic noises in conclusion. However, a limitation is also on the dependence on clean observations present in the dataset when some elements are corrupted.
The authors do not discuss any negative sociatal impact of their work.

---

> ### Author Rebuttal · Authors · 2024-08-07
>
> We appreciate your insightful comments. We respond to each comment as follows and sincerely hope that our responses could properly address your concerns. If so, we would deeply appreciate it if you could raise your score. If not, please let us know your further concerns, and we will continue actively responding to the comments and enhancing our submission.
> - **Q1**. (1) The paper introduces LQ (θi), LV(ψ), Lπ(ϕ) in Eq 4, 5 and 6 but does not discuss how they relate to Dθ. Is θ for learning Q function and D same? (2) How is the final policy extracted after learning Dθ.
>   - **A1**. We apologize for any confusion caused by unclear descriptions in our main text.
>     1. Relation between $Q$, $D$, and $\theta$.
>         - We use **the same $\theta$** to learn $D$ and then estimate $Q$. Specifically, following [1], we have
>         $$Q_{\theta_i}(s,a) = \sum_{n=1}^N D_{\theta_i}^{\tau_n}(s,a,r).$$
>           See Lines 167-172 of the main text for more details regarding the notations.
>
>           Similarly, we also have
>           $$V_{\psi}(s) = \sum_{n=1}^N Z_{\psi}^{\tau_n}(s).$$
>         - With the relation between $Q$ and $D$, $V$ and $Z$, we use Eq. (4) to derive $\delta$ in Eq. (12) and Eq. (5) to derive Eq. (13) in our main text.
>     2. Policy Learning.
>         - Following the weighted imitation learning in RIQL and IQL, we use Eq. (6) in our main text to learn the policy. Details for notations are shown in Lines 122-123 of the main text.
>
>   [1] Implicit quantile networks for distributional reinforcement learning.
> - **Q2**. The paper relies on many hyperparameters.
>   - **A2**. Compared to RIQL, one of SOTA methods, our approach TRACER only introduces additional hyperparameters associated with the approximation of action-value distributions: (1) the number $N$ of samples $\tau$; (2) a linear decay parameter $\beta$ used to trade off the losses $\mathcal{L}\_{\text{first}}$ and $\mathcal{L}\_{\text{second}}$.
> - **Q3**. What kind of kernel is used in Gaussian distribution of Eq 10?
>   - **A3**. We do **not** use a Gaussian kernel in Eq. (10). See the explanation of Eq. (10) in A2 of our global response for more details.
> - **Q4**. Is there a upper bound on what amount of corruption the method can handle?
>   - **A4**. Yes. Please refer to A4 in our global response.
> - **Q5**. While authors consider individual corruptions, data in the real world can have simultaneous corruptions.
>   - **A5**. We specifically design TRACER to handle the challenging simultaneous corruptions. Results in Table 1 of the main text show that TRACER outperforms several SOTAs in **all tasks under simultaneous corruptions**, achieving an average gain of ${\bf +21.1\\%}$. Note that "mixed corruptions" in the main text refers to simultaneous corruptions (see Line 233 in the main text).
> - **Q6**. The paper introduces $q^{\pi}$ in Eq 18. Is this the same as action value distribution?
>   - **A6**. No. The notation $q^\pi$ in Eq. (18) of the Appendix denotes the value distribution, consistent with $p(\cdot|s)$ used in Line 170 of the main text, which is an expactation of action value distribution $\mathbb{E}_{a\sim \pi, r\sim \rho} [p\_\theta(\cdot|s,a,r)]$.
> - **Q7**. RIQL already proves the robustness of IQL policy against data corruption. What is the objective of Theorem A.3?
>   - **A7**. Please refer to A4 in our global response.
> - **Q8**. Providing a table summarizing the notations in appendix will enhance readability.
>   - **A8**. Thanks for the insightful comment. We will include this table in our revision.
> - **Q9**. How relevant is the reward corruption given offline RL already robust to noise in the reward function?
>   - **A9**. While the offline RL is robust to small-scale random reward corruptions [1], it tends to struggle with large-scale reward corruptions (see Page 21 in RIQL).
>
>   [1] Survival instinct in offline reinforcement learning.
> - **Q10**. Do the authors assume that complete traces of the system are available?
>   - **A10**. No. We employ the commonly used offline RL setting [1], where the offline dataset consists of shuffled tuples and agents only use individual tuples rather than complete traces.
>
>   [1] Off-policy deep reinforcement learning without exploration.
> - **Q11**. In situations of dynamic corruption these traces might be different than the system. For example, $s_t$ in the corrupted dataset may not be observed in the real system. How are such cases handled?
>   - **A11**. Thanks for the insightful comment. In a scenario where an element (e.g., state) may be corrupted, TRACER captures uncertainty by using (1) other elements and (2) correlations between all elements and action values (see Lines 53-60 of the main text).
> - **Q12**. What amount of clean data needs to be present in the dataset for this method to work?
>   - **A12**. Please refer to A4 in our global response. Results in Table G4 show that while TRACER is robust to simultaneous corruptions, its performance depends on the extent of corrupted data it encounters.
> - **Q13**. What is the reason for not using projected gradient descent for reward corruptions?
>   - **A13**. Following RIQL, the objective of adversarial reward corruption is $\hat{r} = \min_{\hat{r} \in \mathbb{B}(r, \epsilon)} \hat{r} + \gamma \mathbb{E}[Q(s', a')]$. Here $\mathbb{B}(r, \epsilon)=\\{\hat{r}\mid |\hat{r} - r| \leq (1+\epsilon)\cdot r_{\max} \\}$ regularizes the maximum distance for rewards. Thus, we can directly compute $\hat{r} =-\epsilon\times r$ without using projected gradient descent.
> - **Q14**. Could the author provide some understanding of why the method outperforms RIQL under certain settings and why it does not in some cases?
>   - **A14**. Please refer to A3 in our global response.
> - **Q15**. Limitations.
>   - **A15**. We look forward to developing corruption-robust offline RL with large language models, introducing the prior knowledge of clean data against large-scale or near-total corrupted data. Moreover, we plan to add potential negative societal impacts in our revision.

---

> > ### Author Response · Authors · 2024-08-08
> > **Table of Notations used in Appendix for Q8.**
> >
> > For Q8, we provide the detailed table summarizing the notations used in our Appendix. See Table G8 as follows.
> >
> > Table G8. Notations in our Appendix.
> >
> > | Notations used in our Appendix | Descriptions                                                 |
> > | ------------------------------ | ------------------------------------------------------------ |
> > | $\mathbb{\zeta}$               | Cumulative corruption level.                                 |
> > | $\mathbb{\zeta}_i$             | Metric quantifying the corruption level of rewards and next states (transition dynamics). |
> > | $\mathbb{\zeta}_i^{'}$         | Metric quantifying the corruption level of states and actions. |
> > | $\pi_{b}(\cdot\|s)$            | The behavior policy that is used to collect clean data.      |
> > | $\pi_{\mathcal{B}}(\cdot\|s)$  | The behavior policy that is used to collect corrupted data.  |
> > | $\pi_{E}(\cdot\|s)$            | The policy that we want to learn under clean data.           |
> > | $\tilde{\pi}_{E}(\cdot\|s)$    | The policy that we are learning under corrupted data.        |
> > | $\pi_{\text{IQL}}(\cdot\|s)$          | The learned policy using IQL's weighted imitation learning under clean data. |
> > | $\tilde{\pi}_{\text{IQL}}(\cdot\|s)$  | The learned policy using IQL's weighted imitation learning under corrupted data. |
> > | $d^{\pi}(s,a)$                 | The probability density function associated with policy $\pi$ at state $s$ and action $a$. |
> > | $W_1(p,q)$                     | The Wasserstein-1 distance that measures the difference between distributions $p$ and $q$. |
> > | $q^{\pi}(\cdot \| s)$            | The value distribution of the policy $\pi$.                  |
> > | $Z^{\pi}(s)$                   | The random variable of the value distribution $q^{\pi}(\cdot\|s)$. |
> > | $\epsilon_1$                   | The KL divergence between policies $\pi_{E}$ and $\pi_{\text{IQL}}$, representing standard imitation error under clean data. |
> > | $\epsilon_2$                   | The KL divergence between policies $\tilde{\pi}\_{E}$ and $\tilde{\pi}\_{\text{IQL}}$, representing the standard imitation error under corrupted data. |
> > | $\hat{\varsigma}_n$            | The midpoint of the action-value distribution $p_{\theta}$.        |

---

> > > ### Comment · Reviewer_QZTE · 2024-08-09
> > > **Response to rebuttal**
> > >
> > > Thank you for your efforts in answering the questions and conducting additional experiments. I specifically like Fig G1 which helps in understanding the proposed methodology. I still have the following concerns:
> > >
> > > 1. Theorem A.3 is unclear to me since it is derived in terms of $\tilde \pi_{iql}$ which as per my understand is iql learned on corrupted data. How does this prove the upper bound for TRACER? How do you equate $Z^\pi(s)$ with $D^\pi(s, a)$ is steps 24, 25 and 26. And what is $D^\pi(s, a)$ as previously D is defined as D(s, a, r)?
> > > 2. The authors in Appendix B state we randomly select c% of the transitions each time for corruption. I am interested in knowing what is the value of c experimentally and what the effect on performance for varying c.

---

> ### Author Response · Authors · 2024-08-09
> **Response to Additional Comments**
>
> Thank you for your valuable comments. We respond to each of your comments as follows.
>
> **Q16**. (1) How does $\tilde{\pi}_{\text{IQL}}$ help to prove the upper bound for TRACER?  (2) How do you equate $Z^\pi(s)$ with $D^\pi (s,a)$ in steps 24, 25, and 26 in Theorem A.3? (3) What is $D^\pi (s,a)$ as previously D is defined as D(s, a, r)?
>   - **A16**.
>     - For (1), as TRACER directly applies the weighted imitation learning technique from IQL to learn the policy, we can use $\tilde{\pi}_{\text{IQL}}$ as the policy learned by TRACER under data corruptions, akin to RIQL.
>     In Theorem A.3 of the Appendix, the major difference between TRACER and IQL/RIQL is that TRACER uses the action-value and value distributions rather than the action-value and value functions in IQL/RIQL. Therefore, we further prove an upper bound on the difference in value distributions of TRACER to show its robustness.
>     - For (2), we apologize for the missing reference in steps 24, 25, and 26 of Theorem A.3 (see Lemma 6.1 of [1]). We follow [1] to provide the detailed derivation below.
>       - For any $\pi'$ and $\pi$, we have
>   $$\begin{align}
> Z^{\tilde{\pi}}(s) &= \sum_{t=0}^\infty \gamma^t \mathbb{E}\_{(S_t,A_t)\sim P,\tilde{\pi}} \left[R(S_t, A_t) | S_0 = s\right]\\\\
> &= \sum_{t=0}^\infty \gamma^t \mathbb{E}_{(S_t,A_t)\sim P,\tilde{\pi}} \left[R(S_t, A_t)+Z^{\pi}(S_t)-Z^{\pi}(S_t) | S_0 = s\right]\\\\
> &= \sum\_{t=0}^\infty \gamma^t \mathbb{E}\_{(S\_t,A\_t,S\_{t+1}) \sim P,\tilde{\pi}} \left[R(S\_t, A\_t)+\gamma Z^{\pi}(S\_{t+1})-Z^{\pi}(S\_t) | S\_0 = s\right] + Z^{\pi}(s)\\\\
> &= Z^{\pi}(s)+\sum\_{t=0}^\infty \gamma^t \mathbb{E}\_{(S_t,A_t)\sim P,\tilde{\pi}} \left[D^{\pi}(S\_t,A\_t,R\_t)-Z^{\pi}(S\_t) | S\_0 = s\right]\\\\
> &= Z^{\pi}(s)+ \frac{1}{1-\gamma} \mathbb{E}\_{(s,a)\sim d^{\tilde{\pi}},\tilde{\pi}} \left[D^{\pi}(s,a,r)-Z^{\pi}(s)\right].
> \end{align}$$
>
>         Thus, we can derive the step 25 from the step 24.
>     - For (3), we apologize for any confusion caused by the notations. We will replace the notation $D^\pi (s,a)$ with $D^\pi (s,a,r)$ in Theorem A.3 of our revision.
>
>     [1] Approximately optimal approximate reinforcement learning.
>
> **Q17**. (1) What is the value of $c$ we use in experiments? (2) What is the effect on performance for varying $c$?
>   - **A17**.
>     - For (1):
>       - For each experiment with a random seed under individual corruptions in Tables 2 and 3 of our main text, we randomly select $c\\% = 30\\%$ of transitions from the offline dataset. Within these selected transitions, we replace one element per transition with corrupted element.
>       - In Table 1 with simultaneous corruptions, we also apply $c\\% = 30\\%$ but extend the corruption process across all four elements of the offline dataset. Specifically, we randomly select $30\\%$ of transitions and corrupt one element in each selected transition. Then, we repeat this step four times until all elements are corrupted. Therefore, approximately $76.0\\%$ of data in the offline dataset is corrupted, calculated as $1 - (1 - c)^4$.
>
>         In Table G4 of our attached PDF of the global response, we evaluate TRACER using different $c \\%$, including $10\\%, 20\\%, 30\\%, 40\\%,$ and $50\\%$. These rates correspond to approximately $34.4\\%, 59.0\\%, 76.0\\%, 87.0\\%$, and $93.8\\%$ of the data being corrupted.
>     - For (2):
>       - Results in Table G4 show that while TRACER is robust to simultaneous corruptions and significantly outperforms RIQL, its performance depends on the extent of corrupted data it encounters, degrading with increased data corruptions.
>
> We hope our responses adequately address your concerns.  If you have further concerns, please let us know and we will continue actively responding to your comments, enhancing our submission. We would deeply appreciate it if you could raise your score based on these revisions.

---

> > ### Comment · Reviewer_QZTE · 2024-08-10
> > **Response**
> >
> > Thank you for addressing my concerns. I have raised my score. I would urge the authors to include the notation description, additional results and clear explanation of Theorem A3 in the revised version.

---

> > > ### Author Response · Authors · 2024-08-10
> > > **Thanks for your kind support and for helping us improve the paper!**
> > >
> > > Dear Reviewer QZTE,
> > >
> > > Thanks for your kind support and for helping us improve the paper! We are committed to including all these additional results, notation descriptions, and detailed explanations and will incorporate your suggestions in the final version, if accepted.
> > >
> > > Thank you again for your valuable comments and guidance.
> > >
> > > Best,
> > >
> > > Authors

---

> ### Author Response · Authors · 2024-08-11
> **Thanks again for your continued support!**
>
> Dear Reviewer QZTE,
>
> Thanks again for your continued support and the further improvements you have suggested for our manuscript. We sincerely appreciate your insightful feedback, which has significantly helped us in refining the explanations and enhancing the theoretical derivations. We remain committed to incorporating all your valuable suggestions into the final version, if accepted.
>
> We are deeply grateful for your thoughtful comments and for the confidence you have shown in our work by raising your score.
>
> Best,
>
> Authors

---

### Official Review · Reviewer_V2Jx · 2024-07-11

**Soundness:** 3
**Presentation:** 3
**Contribution:** 3
**Rating:** 6
**Confidence:** 4

**Summary:**

This paper seeks to conduct reinforcement learning from corrupted offline data. More specifically, they propose the TRACER algorithm, which uses bayesian inference to calculate the uncertainty in estimating the action-value function. The authors conduct experiments with diverse corruptions on CARLA and Mujoco environments.

**Strengths:**

- This paper considers an important problem.
- Their proposed algorithm is interesting and performs well.

**Weaknesses:**

- The contribution compared to other baselines (RIQL) is not really clear. In the introduction the authors claim that their method can handle simultaneous corruptions and RIQL cannot. However no experiments are done in this setting.
- The experimental settings are fairly limited: only three Mujoco environments are considered and one carla experiment.
- Different perturbations levels are not shown.
- The authors do not discuss the computational cost of their method.

**Questions:**

- Can you clarify the contribution of TRACER compared to RIQL? It seems like it is an orthogonal approach, but the two algorithms are not really compared in detail.
- Can you show the hyperparameter tuning results? Is TRACER stable under different hyperparameter settings? How does it compare to RIQL in this regard?
- How does the computational cost of TRACER compare to the other baselines?

**Limitations:**

The authors do discuss their method's limitations.

---

> ### Author Rebuttal · Authors · 2024-08-07
>
> We appreciate the reviewer's insightful and constructive comments and suggestions. We respond to each comment as follows and sincerely hope that our responses could properly address your concerns. If so, we would deeply appreciate it if you could raise your score. If not, please let us know your further concerns, and we will continue actively responding to the comments and enhancing our submission.
> - **Q1**. No experiments for simultaneous corruptions.
>   - **A1**. We apologize for the confusion caused by unclear descriptions in our main text. Specifically, the experiments presented in Table 1 on Page 7 use the simultaneous corruptions. The term "**mixed corruptions**" in the caption of Table 1 refers to the simultaneous corruptions (see the explanation on Line 233 of Section 4 in the main text). In our revision, we will replace "mixed corruptions" with "simultaneous corruptions" to avoid any ambiguity.
> - **Q2**. The contribution compared to other baselines (RIQL) is not really clear.
>   - **A2**. Thanks for the kind and insightful comment.
>     - **Relation between TRACER and RIQL**. Our approach TRACER applies the weighted imitation learning in RIQL for policy improvement (see Eq. (6) on Page 4 of the main text). This allows us to build and expand upon RIQL.
>     - **Contributions of TRACER**. We present TRACER's major contributions compared to corruption-robust offline RL methods [1,2] (especially RIQL):
>       1. To the best of our knowledge, TRACER introduces Bayesian inference into corruption-robust offline RL for **the first time**. Thus, it can capture uncertainty caused by diverse corrupted data to **simultaneously** handle the corruptions, unlike other corruption-robust offline RL methods that primarily focus on individual corruptions.
>       2. TRACER can **distinguish corrupted data from clean data** using an entropy-based uncertainty measure. Thus, TRACER can regulate the loss associated with corrupted data to reduce its influence. In contrast, existing corruption-robust offline RL methods lack the capability to identify which data is corrupt.
>       3. Based on Tables 1, 2, and 3 of the main text, TRACER significantly outperforms existing corruption-robust offline RL methods across a range of **both individual and simultaneous** data corruptions.
>
>   [1] Corruption-robust offline reinforcement learning with general function approximation. NeurIPS 2023.
>
>   [2] Towards robust offline reinforcement learning under diverse data corruption. ICLR 2023.
> - **Q3**. The experimental settings are fairly limited.
>   - **A3**. Thanks for your insightful comment.
>     - Our experiments using Mujoco and CARLA datasets are consistent with standard practices in corruption-robust offline RL. Existing methods [1,2] often select these datasets to assess their effectiveness.
>     - Based on the corruption-robust offline RL methods, we further conduct experiments on two AntMaze datasets and two additional Mujoco datasets, presenting results under random simultaneous corruptions in Table G5 of the PDF of our global response.
>       - **Settings**. Each result represents the mean and standard error over four random seeds and 100 episodes in clean environments. For each experiment, the methods train agents using batch sizes of 64 for 3000 epochs. Building upon RIQL, we apply the experiment settings as follows.
>         1. For the two Mujoco datasets, we use a corruption rate of $c=0.3$ and scale of $\epsilon=1.0$. Note that simultaneous corruptions with $c=0.3$ implies that approximately $76.0\\%$ of the data is corrupted.
>         2. For the two AntMaze datasets, we use the corruption rate of 0.2, corruption scales for observation (0.3), action (1.0), reward (30.0), and dynamics (0.3).
>       - **Results**. The results in Table G5 show that TRACER significantly **outperforms** other methods in **all these tasks** with the aforementioned AntMaze and Mujoco datasets.
>
>   [1] Corruption-robust offline reinforcement learning with general function approximation. NeurIPS 2023.
>
>   [2] Towards robust offline reinforcement learning under diverse data corruption. ICLR 2023.
> - **Q4**. Different perturbation/corruption levels are not shown.
>   - **A4**. Thanks for the kind and insightful comment.
>     - **Setting**. Building upon RIQL, we extend our experiments to include Mujoco datasets with different corruption levels, using different corruption rates and scales. We report the average scores and standard errors over four random seeds in Figure G2 of the PDF of our global response, using batch sizes of 256.
>     - **Results**. Figure G2 in the PDF shows that TRACER significantly outperforms other algorithms in **all tasks** under **random simultaneous corruptions**, achieving an average score improvement of ${\bf +33.6\\%}$.
> - **Q5**. Computational cost of TRACER.
>   - **A5**. Please refer to A1 in our global response.
> - **Q6**. Can you show the hyperparameter tuning results?
>   - **A6**. Thank you for the meaningful comment.
>     - **Setting**. We conduct hyperparameter tuning experiments for both TRACER and RIQL, varying values of $\kappa$ in the Huber loss and $\alpha$ for action-value functions. We report results in Tables G6 and G7 of the PDF of our global response. Note that except the first column in Table G6, we use batch sizes of 64 and a learning rate of 0.0003 over four random seeds on Hopper task under adversarial simultaneous corruptions.
>     - **Results**. Tables G6 and G7 reveal that TRACER is more stable than RIQL and consistently outperforms RIQL, achieving an average performance gain of ${\bf +43.3\\%}$ under adversarial simultaneous corruptions.
>
>       Moreover, we find that both TRACER and RIQL reach their respective highest performance at $\kappa=0.1$ and $\alpha=0.25$, and TRACER still achieves a substantial performance gain of ${\bf +24.7 \\%}$ compared to RIQL.

---

> > ### Author Response · Authors · 2024-08-10
> > **Thanks for your kind support and for helping us improve the paper!**
> >
> > Dear Reviewer V2Jx,
> >
> > Thanks for your kind support and for helping us improve the paper! We are committed to including all these additional results and detailed explanations and will incorporate your suggestions in the final version, if accepted.
> >
> > Thank you again for your valuable comments and guidance.
> >
> > Best,
> >
> > Authors

---

> > > ### Comment · Reviewer_V2Jx · 2024-08-10
> > > **Thank you for the discussion**
> > >
> > > Thank you for the discussion and the extra experiments. My concerns are addressed and I will raise my score.

---

### Author Rebuttal · Authors · 2024-08-07

# Global Response
We would like to thank reviewers for their insightful comments. We respond to the collective feedback below and hope that our responses could adequately address these general concerns. If so, we would deeply appreciate it if reviewers could raise the score. If not, please let us know the further concerns, and we will continue to refine our submission in response to the comments.
- **Q1** for **#R V2Jx and wmJN**. Computational cost comparison.
  - **A1**. We provided the average training duration of our approach TRACER for Halfcheetah, Walker2d, and Hopper in Section B.5 of our Appendix.
  - To compare the computational cost, we report the average epoch time on Hopper in Table G1 of the PDF, where results of baselines (including DT [3]) are from [1]. The formula for computational cost is:
    $$\frac{\text{avg epoch time of RIQL in [1]}}{\text{avg epoch time we run RIQL}} \times \text{avg epoch time we run TRACER}.$$
    Note that TRACER requires a long epoch time due to two main reasons:
    1. Unlike RIQL and IQL, which learn one-dimensional action-value functions, TRACER generates multiple samples for the estimation of action-value distributions. Following [2], we generate 32 samples of action values for each state-action pair.
    2. TRACER uses states, actions, and rewards as observations to update models, estimating the posterior of action-value distributions.

    In the future work, we plan to improve TRACER’s computational efficiency by optimizing codes to estimate the posterior in parallel using various observations.

  [1] Towards robust offline reinforcement learning under diverse data corruption.

  [2] Implicit quantile networks for distributional reinforcement learning.

  [3] Decision transformer: Reinforcement learning via sequence modeling.
- **Q2** for **#R QZTE and wmJN**. Explanation for Eqs. (10) and (11).
  - **A2**. We apologize for any confusion caused by unclear descriptions for Eqs. (10) and (11).
    - The goal of **Eq. (10)** is to estimate $\pi_{\mathcal{B}}(A|S)$, $\rho_{\mathcal{B}}(R|S,A)$, and $p_{\mathcal{B}}(S)$ using $p_{\varphi_a}(A|D_\theta,S,R,S')$, $p_{\varphi_r}(R|D_\theta,S,A)$, and $p_{\varphi_s}(S|D_\theta,A,R)$, respectively. We model all these distributions as Gaussian distributions, and use the mean $\mu_{\varphi}$ and standard deviation $\Sigma_{\varphi}$ to represent the corresponding $p_{\varphi}$. **For implementation**, we employ MLPs to output each $(\mu_{\varphi},\Sigma_{\varphi_r})$ using the corresponding conditions of $p_{\varphi}$. Then, based on the KL divergence between two Gaussian distributions, we can derive Eq. (10).
    - The goal of **Eq. (11)** is to maximize the likelihoods of $D_\theta$ given samples $\hat{s} \sim p_{\varphi_s}$, $\hat{a} \sim p_{\varphi_a}$, or $\hat{r} \sim p_{\varphi_r}$. Thus, with $(s,a,r)\sim \mathcal{B}$, we propose minimizing the distance between $D_\theta (\hat{s},a,r)$ and $D(s,a,r)$, $D_\theta (s,\hat{a},r)$ and $D(s,a,r)$, and $D_\theta (s,a,\hat{r})$ and $D(s,a,r)$, where $\hat{s} \sim p_{\varphi_s}$, $\hat{a} \sim p_{\varphi_a}$, and $\hat{r} \sim p_{\varphi_r}$, thus deriving Eq. (11).
- **Q3** for **#R QZTE and wmJN**. Results comparison between TRACER and RIQL.
  - **A3**. Thanks for the insightful comment.
  - **Simultaneous Corruptions**. Our method TRACER is specifically designed to address simultaneous corruptions for robustness. Results in Table 1 of the main text show that TRACER outperforms RIQL across **all tasks** under simultaneous corruptions, achieving an average gain of ${\bf +21.1\\%}$.

    This is because that TRACER captures uncertainty via offline data against simultaneous corruptions. Thus, it can use uncertainty to distinguish corrupted data from clean data and then reduce the influence of corrupted data.
  - **Individual Corruptions**. In Tables 2 and 3 of the main text, we adhered to commonly used settings for individual corruptions in corruption-robust offline RL. We directly followed hyperparameters from RIQL (i.e., $\kappa$ for huber loss, the ensemble number $K$, and $\alpha$ in action-value functions). Results show that TRACER outperforms RIQL in **18 out of 24** settings, demonstrating its robustness even when aligned with RIQL’s hyperparameters.

    Further, we explore hyperparameter tuning, specifically of $\kappa$, on Hopper to improve TRACER's performance. This results in TRACER outperforming RIQL in **7 out of 8** settings on Hopper, up from 5 settings (see Tables G2 and G3 in the PDF). The further improvement highlights TRACER’s potential to achieve greater performance gains.
    - Based on Table G2, we find that TRACER requires low $\kappa$ in Huber loss, using L1 loss for large errors. Thus, TRACER can linearly penalize the corrupted data and reduce its influence on the overall model fit.
- **Q4** for **#R QZTE**. (1) RIQL already proves the robustness of IQL. What is the objective of Theorem A.3? (2) Is there a upper bound on what amount of corruption the method will be able to handle?
  - **A4**.
    1. Theorem A.3 in our Appendix builds upon RIQL and extends it to TRACER. Specifically, RIQL first proves an upper bound on **the distance in value functions** that IQL can learn under clean and corrupted data. Then, we provide Theorem A.3 to prove an upper bound on **the difference in value distributions** that TRACER can learn under clean and corrupted data. This theorem not only supports the robustness claims of TRACER but also provides a guarantee of how TRACER's performance degrades with increased data corruptions.
    2. Yes. Theorem A.3 shows that the higher the scale of corrupted data, the greater the difference in action-value distributions and the lower the TRACER's performance. We also evaluate TRACER across various corruption levels. Table G4 in the PDF shows that while TRACER is robust to simultaneous corruptions, its scores depend on the extent of corrupted data it encounters.

---

### Decision · Program_Chairs · 2024-09-25

**Decision:**

Accept (poster)

**Comment:**

The reviewers agreed that the paper addresses an important problem of robust reinforcement learning under corrupted data, proposes an interesting method based on Bayesian inference, and the extensive empirical evaluation demonstrates the effectiveness of the proposed method. However, the reviewers also raised several concerns and questions in their initial reviews. We want to thank the authors for their responses and active engagement during the discussion phase. The reviewers appreciated the responses, which helped in answering their key questions. The reviewers have an overall positive assessment of the paper, and there is a consensus for acceptance. The reviewers have provided detailed feedback, and we strongly encourage the authors to incorporate this feedback when preparing the final version of the paper.